# Countries most exposed to individual and concurrent extremes, and near-permanent extreme conditions at different global warming levels

Fulden Batibeniz[1], Mathias Hauser[1], Sonia I. Seneviratne[1]

[1]Institute for Atmospheric and Climate Science, Department of Environmental Systems Science, ETH Zurich, Zurich, Switzerland

*Correspondence to*: Fulden Batibeniz (fulden.batibeniz@env.ethz.ch)

**Abstract.** It is now certain that human-induced climate change is increasing the incidence of extreme temperature, precipitation, and drought events globally. A critical aspect of these extremes is their potential concurrency that can result in substantial impacts on society and environmental systems. Therefore, quantifying concurrent extremes in current and projected climate is necessary to take measures and adapt to future challenges associated with such conditions. Here we investigate changes in individual and concurrent extremes in multi-model simulations of the 6th phase of the Coupled Model Intercomparison Project (CMIP6) for different global warming levels (GWLs). We focus on the individual and simultaneous occurrence of the extreme events, encompassing heatwaves, droughts, maximum 1-day precipitation (Rx1day) and extreme wind (wind) as well as the compound events heatwave-drought and Rx1day-wind in the pre-industrial period (1850-1900; reference period), for approximately present conditions (1°C of global warming), and at three higher global warming levels (GWLs of +1.5°C, +2°C and +3°C). We focus our analysis on 139 countries and three climatic macro-regions: Northern Mid- and High Latitude Countries (MHC), Subtropical Countries (STC), and Tropical Countries (TRC). We find that, on a global scale, most individual extremes become more frequent and affect more land area for higher GWLs. Changes in frequency of individual heatwaves, droughts, Rx1day and extreme wind with higher GWLs cause shifts in timing and disproportional increases in frequency of concurrent events across different months and different regions. As a result, concurrent occurrences of the investigated extremes become 2.0 to 9.6 times more frequent at +3°C of global warming compared to the pre-industrial period. At +3°C the most dramatic increase is identified for concurrent heatwave-drought events with 9.6 times increase for MHC, a 8.4 times increase for STC, and a 6.8 times increase for TRC compared to the pre-industrial period. By contrast, Rx1day-wind events increased the most in TCR (5.3 times), followed by STC (2.3 times) and MHC (2.0 times) at +3°C with respect to the pre-industrial period. Based on the 2015 population, these frequency changes imply an increase in the number of concurrent heatwave-drought (Rx1day-wind) events per capita for 82% (41%) of countries. Our results also suggest that there are almost no time periods (on average zero or only one month per year) without heatwaves, droughts, Rx1day and extreme wind for 21 countries at +1.5°C of global warming, 37 countries at +2°C and 85 countries at +3°C, compared to 2 countries at 1°C of global warming. This shows that a large number of countries will shift to near-permanent extreme conditions even at global warming levels consistent with the limits of the Paris

Agreement. Given the projected disproportional frequency increases and decreasing non-event months across GWLs, our results strongly emphasize the risks of uncurbed greenhouse gas emissions.

**Plain Language Summary**

We study single and concurrent heatwaves, droughts, precipitation, and wind extremes. Globally, these extremes become more frequent and affect larger land areas under future warming, with several countries experiencing extreme events every single month. Concurrent heatwaves-droughts (precipitation-wind) are projected to increase the most in mid-high latitude countries (tropics). Every mitigation action to avoid further warming will reduce the number of people exposed to extreme weather events.

**1. Introduction**

The socio-economic impacts of individual and concurrent extremes are accelerating with increasing global warming (IPCC, 2021). The intervals between extremes are becoming shorter, which puts vulnerable communities and ecosystems at risk. In addition, while most countries are affected by climate extremes, some economies in global south are more vulnerable than advanced economies in the Northern Hemisphere (Guo et al., 2021). These emerging challenges motivate the need for a

comprehensive analysis of potential changes in exposure to individual and concurrent extremes on the population- and country-level.

Human-induced climate change is exacerbating climate extremes in every region across the globe (Seneviratne et al., 2021). This increase in climate extremes cannot be explained without human influence on the climate system and threatens both developed countries and developing countries. It is noteworthy that low-income and high-population countries have been the

most affected by climate extremes in terms of economic and environmental fatalities during the last two decades (Eckstein et al., 2021). This indicates the inequity between $CO_2$ high-emitter and low-emitter countries when dealing with climate-induced risks and impacts. Our motivation here is to provide a comprehensive assessment of potential changes in population exposure to climate extremes across countries with different climates.

Previous studies typically focus on current and/or projected changes of single extremes (Tebaldi et al., 2006; Orlowsky and

Seneviratne, 2012; Alexander et al., 2006; Westra et al., 2013; Mondal and Mujumdar, 2015; Bao et al., 2017; Alizadeh et al., 2022), whereas recently there has been more attention on compound events - multiple extremes occurring either simultaneously or/and consecutively - due to the rising awareness about their potential amplified impacts (Seneviratne et al., 2010; Mazdiyasni and AghaKouchak, 2015; Forzieri et al., 2016; Zscheischler and Seneviratne, 2017; Vogel et al., 2017; Batibeniz et al., 2020a; Vogel et al., 2020; Saeed et al., 2021; Schwingshackl et al., 2021; Kelebek et al., 2021). The impacts

associated with compound events are expected to be higher than impacts caused by individual extremes. For example, a combination of extreme wind and extreme precipitation can increase the destruction of infrastructure and economic losses. As climate change alters the nature of weather and climate events (extreme or not), compound events composed of these

events are expected to be unprecedented in terms of severity and intensity (Seneviratne et al., 2021). This emerging understanding makes it necessary to quantify the projected changes in the characteristics of both individual and compound events.

A range of obstacles hinders a reliable estimation of the likelihood of compound events. Extreme events are rare by definition and compound extreme events even more so. Additionally, a robust understanding and detailed spatiotemporal information on exposure to multivariate extremes requires high spatiotemporal coverage. This hinders the assessment of observation-based compound events. Therefore, large model ensembles (Champagne et al., 2020; Poschlod et al., 2020; Vogel et al., 2020; Ridder et al., 2021), process-based model simulations (Couasnon et al., 2020) and reanalysis data (Martius et al., 2016) can complement observational data. In particular, multi-GCM ensembles capture the uncertainty in the large-scale climate and can be a useful tool to investigate compound events in current and future climate.

In this study, we investigate the individual occurrences of heatwaves, droughts, extreme precipitation, and extreme wind as well as concurrent heatwave-drought, and extreme precipitation and extreme wind events, all of which can have severe impacts on different sectors. The first combination – heatwave-drought - influences wildfire, crops, natural vegetation, power plants and fisheries (Zscheischler et al., 2020). The second combination - extreme wind and precipitation - can cause storm surges, floodings, and result in the destruction of infrastructure and damage to the economy. Several studies have found that heatwave-drought occurrences have increased in the last four to five decades (Schubert et al., 2014; Mazdiyasni and AghaKouchak, 2015; Sharma and Mujumdar, 2017; Zscheischler and Seneviratne, 2017; Kirono et al., 2017; Zhou and Liu, 2018; Hao et al., 2018; Sarhadi et al., 2018; Manning et al., 2019; Alizadeh et al., 2020; Feng et al., 2020; Kong et al., 2020; Li et al., 2020; Ridder et al., 2020; Mukherjee and Mishra, 2021; Wu et al., 2021) and are projected to increase in the future (Diffenbaugh et al., 2015; Herrera-Estrada and Sheffield, 2017; Sedlmeier et al., 2018; Li et al., 2019). This increase is mostly attributed to the increase in heatwave occurrences (Bevacqua et al., 2022). Indeed, even when droughts alone do not display an increasing tendency, compound occurrences of heatwave and drought events are expected to increase (Sarhadi et al., 2018; Yu and Zhai, 2020). Compound precipitation and wind extremes have also been investigated in the observational period over many regions including the Mediterranean Basin (Raveh-Rubin and Wernli, 2015), Europe (De Luca et al., 2020; Zscheischler et al., 2021), Great Britain (Tilloy et al., 2021), and at the global scale (Martius et al., 2016; Messmer and Simmonds, 2021). However, these studies differ in methodology, time and spatial scale and future changes of precipitation-wind extremes have, to the best of our knowledge, not been covered in the compound event context or not been evaluated together with heatwave-drought events.

Here we analyse changes in frequency and timing of climate-induced individual and concurrent extreme events, as well as the population exposure to these events. It is important to note that for a risk assessment vulnerability would also have to be considered, but this lies beyond the scope of this study. Building on previous work on projected changes in compound extreme events and human exposure (Batibeniz et al., 2020a; Lange et al., 2020; Chen et al., 2020; Mukherjee et al., 2021; Liu et al., 2021; Alizadeh et al., 2022; Das et al., 2022; Shen et al., 2022), we investigate for the first time the human exposure to co-occurring extreme precipitation-wind events, in addition to co-occurring heatwave-drought events and

individual extremes. We do so in a manner consistent with the 6th Assessment Report of the Intergovernmental Panel on Climate Change (IPCC AR6) framework by analysing the projections for different global warming levels (GWLs, +1°C, +1.5°C, +2°C and +3°C) relative to pre-industrial conditions on country and regional scales.


## 2. Data and Methods

### 2.1 Climate model data

We use CMIP6 simulations (Eyring et al., 2016) of 14 climate models to perform individual and concurrent event analysis in the pre-industrial period (1850-1900) and at four GWLs (see below) for the shared socioeconomic pathway (SSP) projection
marking the high end of future forcing pathways (SSP5-8.5) (Jones and O'Neill, 2016, 2020). The SSP5-8.5 experiment represents high mitigation and low adaptation challenges resulting in radiative forcing of 8.5 W/m$^2$ by the end of 2100. Because we present our results at GWLs we do not expect our results to strongly depend on the choice of the scenario (Seneviratne et al., 2016; Seneviratne and Hauser, 2020; Wartenburger et al., 2017). We use the same ensemble member (r1i1p1) of each model. We retrieve daily maximum temperature, precipitation, maximum wind and soil moisture data from
each model and use conservative remapping (Jones, 1999) to regrid them onto a common 2.5° x 2.5° longitude-latitude grid to enable comparison across different models. The full list of models is provided in Table A1.

### 2.2. Population Counts

In the population exposure analysis, we use gridded population counts retrieved from Gridded Population of the World version 4 (GPWv4) data set (Center for International Earth Science Information Network - CIESIN - Columbia University,
2018). The GPWv4 dataset provides population distributions at various grid resolutions. For our analysis, we use the 1° resolution data which we transform into 2.5° grid resolution to match the resolution of the climate data. GPWv4 data is available for the period from 2000 to 2020 at 5-year intervals. However, we only use 2015 population counts in this paper as they are representative of the world population at +1°C of global warming. To investigate the effect of climate change, we keep the population fixed at 2015 levels for approximately 1°C of global warming while allowing the counts of climate
events to change at GWLs. This approach enables us to examine the cause-effect relationship between increasing temperatures and projected changes in extreme events. Furthermore, using climate change projections and population distributions in combination allows us to investigate changes in the exposure to climate extremes at the regional and country levels.

### 2.3. Climate Regions

We focus our analysis on three climatic macro regions: Northern Mid- and High Latitude Countries (MHC), Subtropical Countries (STC), Tropical Countries (TRC) (Figure 1). These climate regions are created by aggregating country polygons.

The assessments are performed and presented both in regional and country scale to emphasize the response of different climatic regions/countries to individual and concurrent extremes. We show results for climate regions in section 3.1-3.2 figures and on country level in section 3.3-3.5 figures.


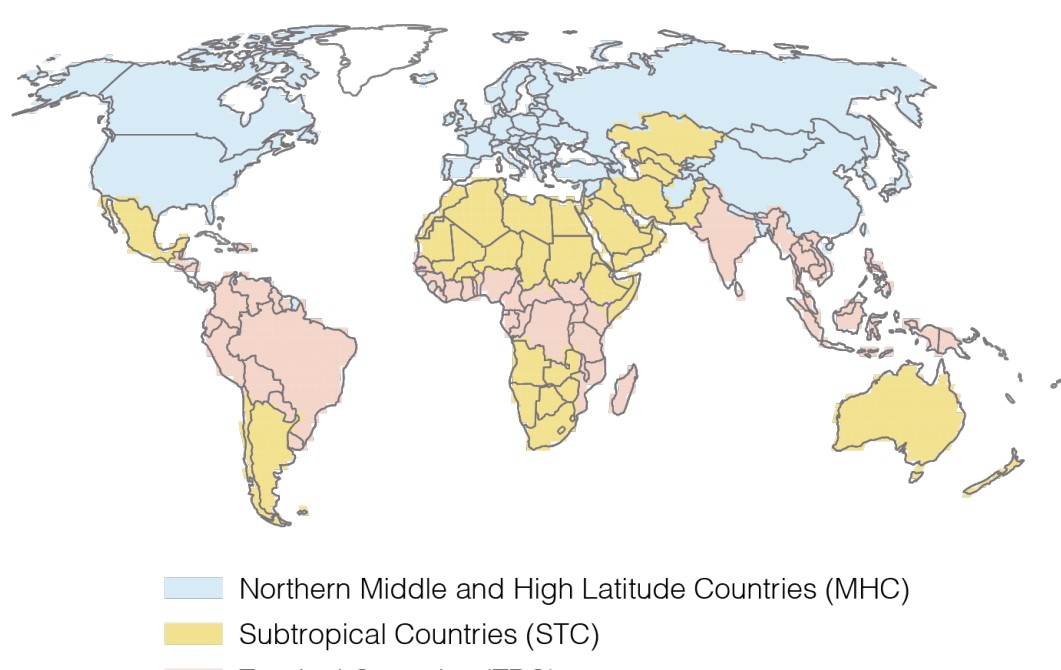

🟦 Northern Middle and High Latitude Countries (MHC)
🟨 Subtropical Countries (STC)
🟧 Tropical Countries (TRC)

**Figure 1.** World map is divided into 3 climatic macro regions: Northern Mid- and High Latitudes Countries (MHC), Subtropical Countries (STC), and Tropical Countries (TRC).

### 2.4. Global temperature and warming level calculation

We perform our analysis considering +1°C, +1.5°C, +2°C and +3°C global warming levels to be consistent with the IPCC AR6 context (Seneviratne et al., 2021). Warming levels are 20-years periods unique to each model due to different climate sensitivity and internal variability. The warming levels are defined as the first 20-year period where global mean temperature anomalies exceed the given temperature (e.g., +2.0°C). We first calculate the annual average global temperature (Figure 2a). Then, we subtract the average global temperature of the pre-industrial period (1850-1900; reference period) from every year

between 1850-2100 and take the 20-years running mean (Figure 2b). The first year a certain anomaly such as +1°C, +1.5°C, +2°C, and +3°C is exceeded is the central year of the warming level period and the warming level period is obtained by subtracting ten and adding nine to the central year (Figure 2b, c; horizontal bars). For example, IPSL-CM6A-LR first exceeds +2°C warming in 2036 so the period selected for this model is 2025-2044 (Figure 2c, red bar). On the other hand, MRI-ESM2-0 reaches +2°C warming in 2040 and the period selected is 2029-2048 (Figure 2c, orange bar).

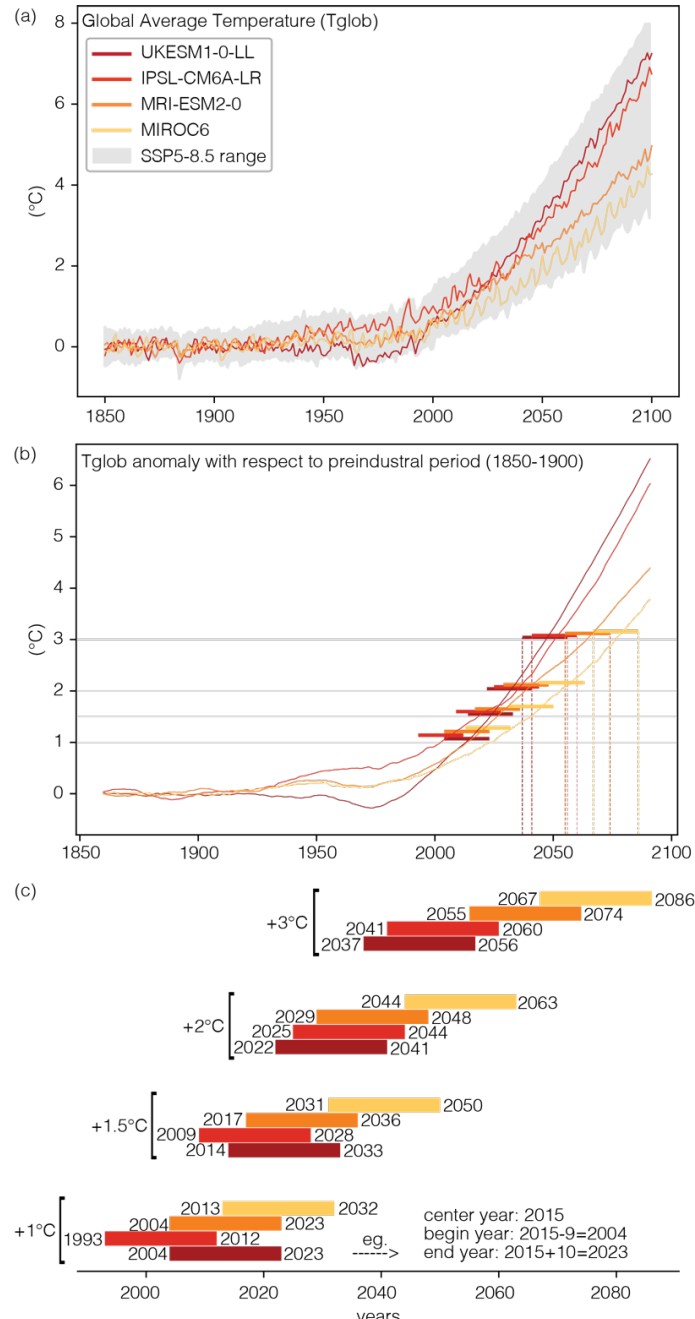


**Figure 2.** Global warming level calculation steps: (a) Global average temperature for four example models under SSP5-8.5 scenario. Coloured lines refer to four models and shaded grey area refers to the spread of temperature variability in all SSP5-8.5 CMIP6 models. (b) 20-year running average of temperature anomaly with respect to pre-industrial period. Horizontal bars represent warming level periods (+1°C, +1.5°C, +2°C, +3°C) for each model and are shifted vertically to ease

understanding. (c) Zoomed version of bars (warming level periods) in (b) to show corresponding years.

## 2.5. Definition of individual events

For our analysis, we calculate heatwaves, drought, heavy precipitation, and extreme wind events empirically. For each model, we define extreme events based on their occurrences below (above) the 10th (90th) percentile during the pre-industrial period with a bootstrap resampling procedure (section 2.7 for details). All the calculations are performed on the 2.5° x 2.5° grid for daily values. The daily events are then aggregated to a monthly timescale, such that a month with one or more daily events is an "event month", and a "non-event month" otherwise.

Heatwave: We use daily maximum temperature to determine heatwave events. We first calculate the 90th percentile for each calendar day using a 31-day moving window over the pre-industrial period with a bootstrap resampling procedure. We then identify a day as a heatwave event if the daily maximum temperature exceeds the daily 90th percentile for at least three consecutive days.

Drought: We compute drought using daily soil moisture data. We use soil moisture to define drought events because it directly represents water availability, in contrast to many other measures (e.g. the standardized precipitation index, SPI) that are based on precipitation scarcity (Seneviratne et al., 2010). We first normalize soil moisture by subtracting the mean of each month and dividing it by its standard deviation over the pre-industrial period. We then compute the 10th percentile for each calendar day using a 31-day moving window over the pre-industrial period as in heatwave calculation. The day is then defined as a drought event if it falls below its 10th percentile.

Rx1day: We use daily precipitation to calculate monthly maximum 1-day precipitation events. We find the maximum 1-day precipitation of each month in the pre-industrial period and define the 90th percentile for each calendar month. Heavy precipitation events are then defined as the days where precipitation is above the monthly threshold.

Extreme wind: We use maximum daily wind speed to calculate extreme wind. For the 90th percentile calculation we use monthly maximum wind speeds in the pre-industrial period. Extreme wind speed days are then defined as days where daily wind speed is above the 90th percentile.

## 2.6. Definition of concurrent events

We define concurrent events as events that occur on the same day in a month and affect the same location. We assess two types of concurrent events: combined heatwave and drought events as well as Rx1day and extreme wind events. Thus, if a specific month experiences two individual events on the same day, it is marked as "event month", for that grid cell and month. For example, if there is a drought event occurring on the same days with a heatwave event regardless of the number of concurrent events, we mark that month as an "event month" otherwise "non-event month".

## 2.7. Bootstrap Resampling procedure

Percentile based indices for climate change detection may create artificial jumps at the beginning and end of the reference period (Zhang et al., 2005). These discontinuities can lead to an artificial frequency increase outside the reference period.

Therefore, we used the bootstrap resampling procedure proposed by Zhang et al. (2005) to overcome this problem. From the 51-year reference period we consecutively excluded one year and included one random year from the remaining years which the thresholds are estimated. The thresholds we found from every iteration is used on the excluded year. Fifty-one thresholds obtained from bootstrap resampling procedures are then averaged and used for the future period. Applying this procedure improved our results in terms of inhomogeneities occurring outside the reference period for heatwave, Rx1day and extreme wind; however, it didn't affect the drought frequencies. Nevertheless, we used this approach to estimate the thresholds of all extreme indices to be consistent methodologically. We refer readers to Zhang et al. (2005) for detailed information about the bootstrap resampling procedure.

## 3. Results

### 3.1 Future changes in individual and concurrent extremes over the climate regions

We illustrate the development of the investigated events with the help of Venn diagrams, which allow us to analyse the frequencies of individual, isolated and concurrent exceedances at the same time. We visualize the individual events by circles and their concurrency by the intersection of these circles. Given two event types A and B, the three numbers on the sets represent the frequencies of isolated first event (A-(A∩B)), concurrent event (A∩B) and isolated second event (B-(A∩B)) in percentage. The two numbers over the sets show the individual event shares of first (A) and second event (B), respectively. The displayed results represent the regional and multi-model mean. The reason we illustrate the mean instead of a median is to avoiding showing different shares from different models for each set. We thereby focus on three continental climate regions (MHL, STC, TRC) for pre-industrial, current (+1°C) and future climate (+1.5°C, +2°C, and +3°C) (Figure 3 a, b).

At the current warming level, the isolated heatwave frequency more than doubled compared to pre-industrial levels in MHC (2.1 times more compared to pre-industrial levels) and STC (2.6), and it quadrupled in TRC (4.4) (Table 1). The event fraction at +3.0°C and the acceleration of the increase across warming levels in isolated heatwave events are the most in TRC (55.3%, 7.5 times more compared to pre-industrial levels). Isolated drought events, on the other hand, tend to decrease for higher GWLs in all regions. This is mostly because, drought events that occur together with heatwave events increase with higher GWLs. Concurrent heatwave and drought events are projected to increase in all climate regions with higher GWLs. At the current GWL, the number of concurrent events is estimated to occur about ~3 times more frequently for MHL and TRP and 4.6 times more frequently for STC compared to the pre-industrial period. The strongest increase across the warming levels occurs for MHC (9.6) and STC (8.4) followed by TCR (6.8). The event fraction at +3.0°C is similar in MHC and STC (24.0% and 23.6%) and greater in TCR (33.2%), however, the proportional increase is the strongest in MHC compared to pre-industrial levels.

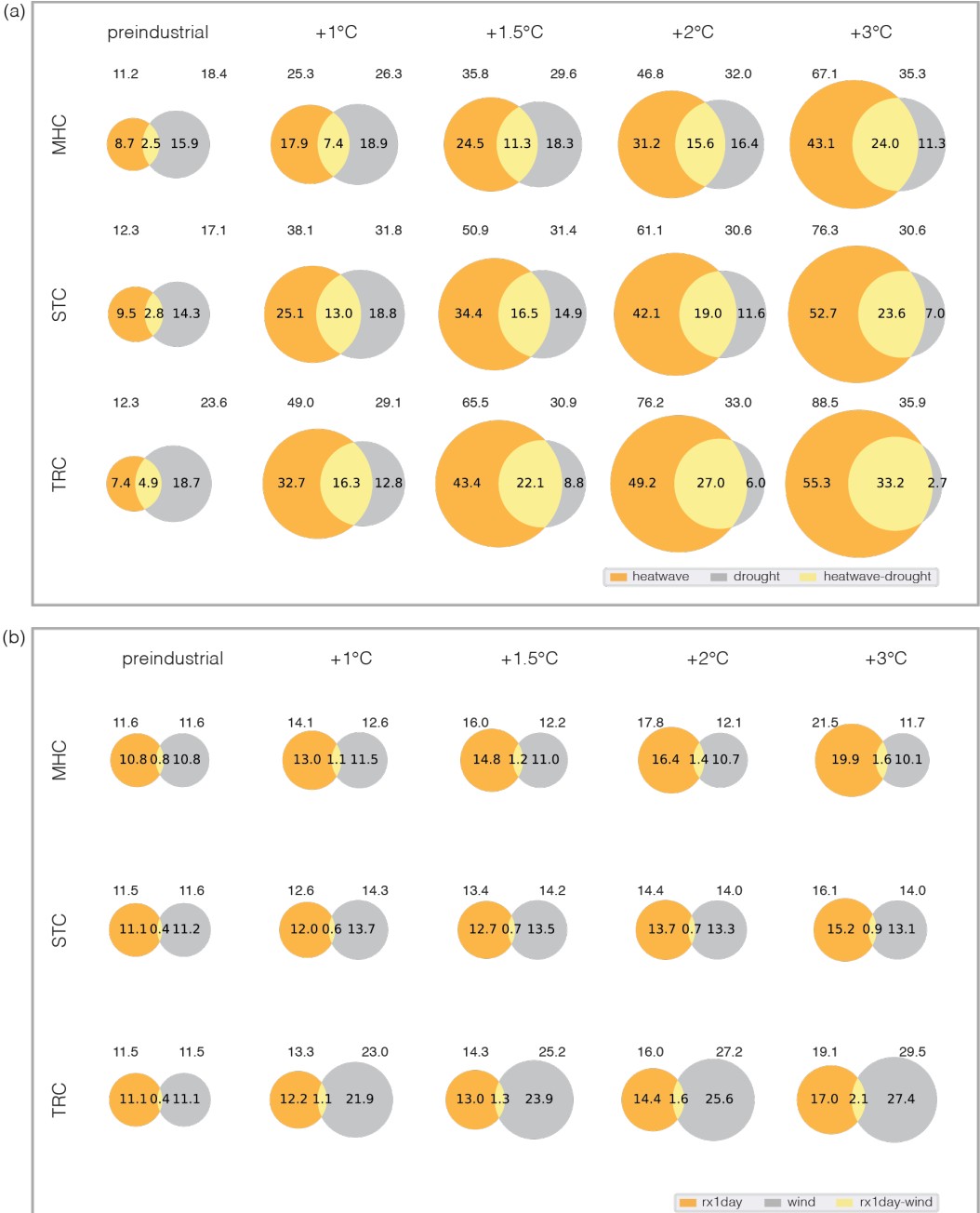

**Figure 3.** Venn diagrams of (a) heatwave-drought and (b) Rx1day-wind storm events at global warming levels. The values show the individual and concurrent frequency of events in MHC, STC and TRC in preindustrial period and at +1°C, +1.5°C, +2°C and +3°C GWLs. Areas of the circles are proportional to the frequencies [%] and represent the multi-model mean. The numbers above the Venn diagrams represent the total share of individual events including the ones occurring during concurrent events.

**Table 1.** Increase in isolated and concurrent events at global warming levels relative to pre-industrial levels.

| | MHC | | | STC | | | TRC | | |
|---|---|---|---|---|---|---|---|---|---|
| | hw | hw-drought | drought | hw | hw-drought | drought | hw | hw-drought | drought |
| +1°C | 2.1 | 3.0 | 1.2 | 2.6 | 4.6 | 1.3 | 4.4 | 3.3 | 0.7 |
| +1.5°C | 2.8 | 4.5 | 1.2 | 3.6 | 5.9 | 1.0 | 5.9 | 4.5 | 0.5 |
| +2°C | 3.6 | 6.2 | 1.0 | 4.4 | 6.8 | 0.8 | 6.6 | 5.5 | 0.3 |
| +3°C | 5.0 | 9.6 | 0.7 | 5.5 | 8.4 | 0.5 | 7.5 | 6.8 | 0.1 |
| | MHC | | | STC | | | TRC | | |
| | Rx1day | Rx1day-wind | wind | Rx1day | Rx1day-wind | wind | Rx1day | Rx1day-wind | wind |
| +1°C | 1.2 | 1.4 | 1.1 | 1.1 | 1.5 | 1.2 | 1.1 | 2.8 | 2.0 |
| +1.5°C | 1.4 | 1.5 | 1.0 | 1.1 | 1.8 | 1.2 | 1.2 | 3.3 | 2.2 |
| +2°C | 1.5 | 1.8 | 1.0 | 1.2 | 1.8 | 1.2 | 1.3 | 4.0 | 2.3 |
| +3°C | 1.8 | 2.0 | 0.9 | 1.4 | 2.3 | 1.2 | 1.5 | 5.3 | 2.5 |

In Figure 3b, we show Rx1day and wind events. The most dramatic increase in isolated and individual Rx1day events is detected in MHC. The frequency of isolated Rx1day events gradually increases across the warming levels by a factor of 1.2, 1.4, 1.5 and 1.8 for +1°C, +1.5°C, +2°C, and +3°C with respect to pre-industrial levels. The hotspot for isolated wind events is TRC. The increase reaches 2.0 times at +1°C GWL and continues to increase to 2.2, 2.3 and 2.5 for +1.5°C, +2°C, and +3°C. For MHL and STC, isolated and individual wind events show an increasing tendency up to +1°C and start to decrease at +1.5°C, +2°C and +3°C. On the other hand, concurrent Rx1day and wind events are already ~1.5 times the pre-industrial levels at 1°C warming and are projected to increase further for 3°C GWL. Even though the percentage of concurrent events is smaller compared to isolated and individual events, the relative increase is larger across warming levels. Concurrent Rx1day-wind event fractions are projected to increase 5.3 times for TRP, 2.3 times for STC and 2.0 times for MHC at +3°C GWL.

## 3.2. Timing of individual and concurrent extremes

To gain further insight info future individual and concurrent extremes across the climate regions, we now focus on their frequency and timing for each calendar month under pre-industrial conditions and at GWLs (Figure 4 and 5). Again, we first consider heatwave and drought events (Figure 4). As expected, heatwaves increase strongly with global warming (Figure 4, top row). At +1°C of global warming, the associated changes are already far beyond the conditions from pre-industrial levels and show further gradual increase across the global warming levels. The increase in heatwaves are heterogenous across months. This unequal distribution leads to much larger increases in some months than suggested by the annual average (Figure 3). The increase is especially inhomogeneous for MHC. At +1°C of global warming, heatwave events occur mostly in summer. However, for higher GWLs there is a sharp increase for most months especially July and August. In STC and TRC, the increase across warming levels is more homogenous, with a slight shift towards June, July, and August in STC.

Due to its less variable structure in time, drought indicates more continuous increase across months for all regions (Figure 4, middle row). The most dramatic increase of drought is observed for summer months in MHC, while STC and TRC show a relatively homogenous increase over the months. Interestingly, STC sees a small decrease in individual drought events in most months for 3°C warming.

The development of concurrent heatwave-drought events is not simply the combination of the individual events (Figure 4, bottom row). They also show a general increase which, however, has some distinct features. The pattern in MHC is especially interesting: the months from June to October indicate a sharp increase, in contrast to the winter months. For STC, the frequency increase is maximum in September for +3°C warming. With higher GWLs there is a shift in timing of the highest values from summer to autumn months. While it is at its highest in July for +1°C of global warming, it is at its highest in August for +3°C of global warming. In the case of TRC, there is also a shift in the timing of the maximum frequency of heatwave-drought events. For +1°C of warming May shows the highest value, whereas for +3°C of warming June stands out. We observe highest frequency increases in summer and autumn months with respect to preindustrial levels.

The Rx1day, wind, and Rx1day-wind events mostly indicate an increase across all months and warming levels (Figure 5). However, in some regions it is not uniform across months. Individual occurrences of Rx1day events are on the rise across the GWLs and regions. At +3°C of global warming, MHC indicates the highest increase in months between October and May. The increase is more homogenous for STC and TRC. Nonetheless, events seem to increase the most in August and September for STC and November for TRC for the highest GWL. Wind extremes vary more compared to Rx1day events across the regions. The most dramatic increase is identified in TRC from June to November. The second most increase in frequency is observed for STC followed by MHC. However, it is interesting to note that for STC while the months between June and September indicate an increase, the rest of the months indicate a decrease with higher GWLs. Additionally, the highest frequency in extreme wind is observed in August. These increases in individual event frequencies leads to a difference among the regions for concurrent Rx1day-wind events. While there is an increase in winter and spring for MHC, there is an increase in July for STC and all months but especially June to October in TRC.

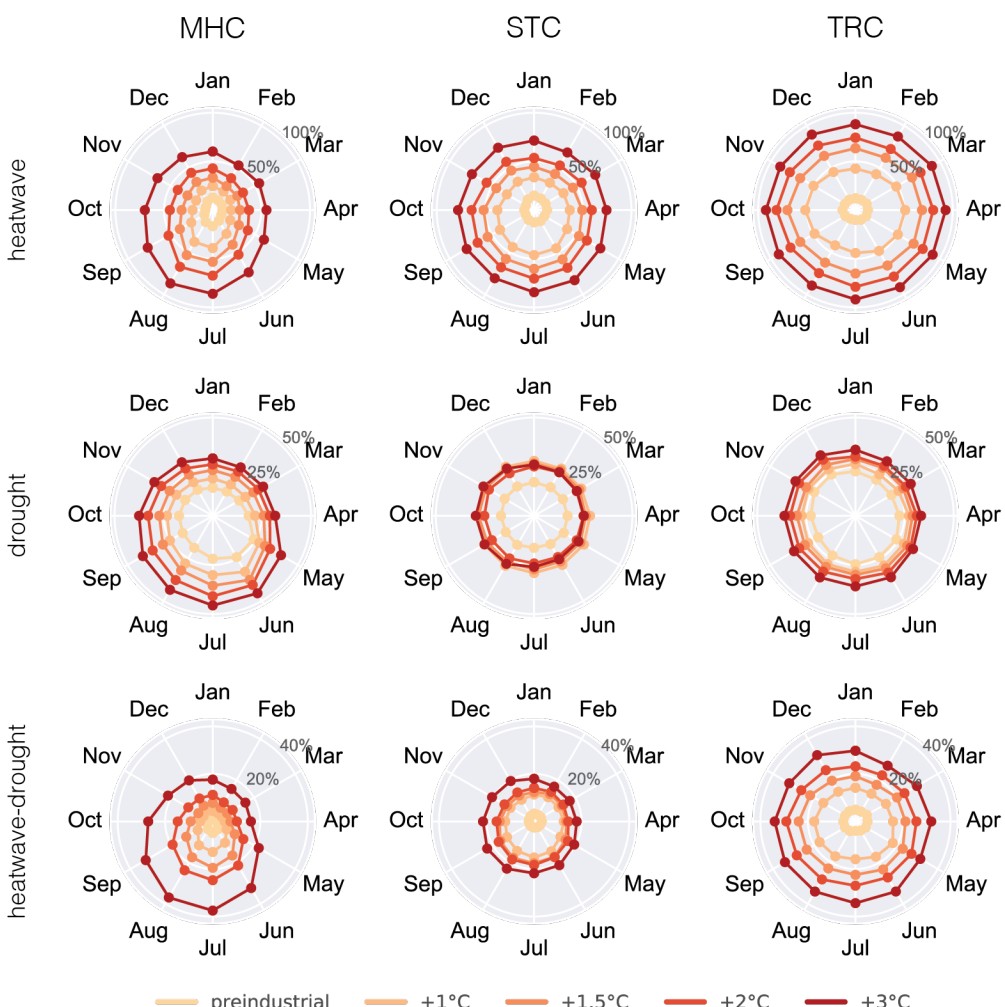

**Figure 4.** Timing and frequency of heatwave, drought, and concurrent heatwave-drought events in MHC, STC and TRC at preindustrial period and at +1°C, +1.5°C, +2°C and +3°C global warming levels in percent.

270

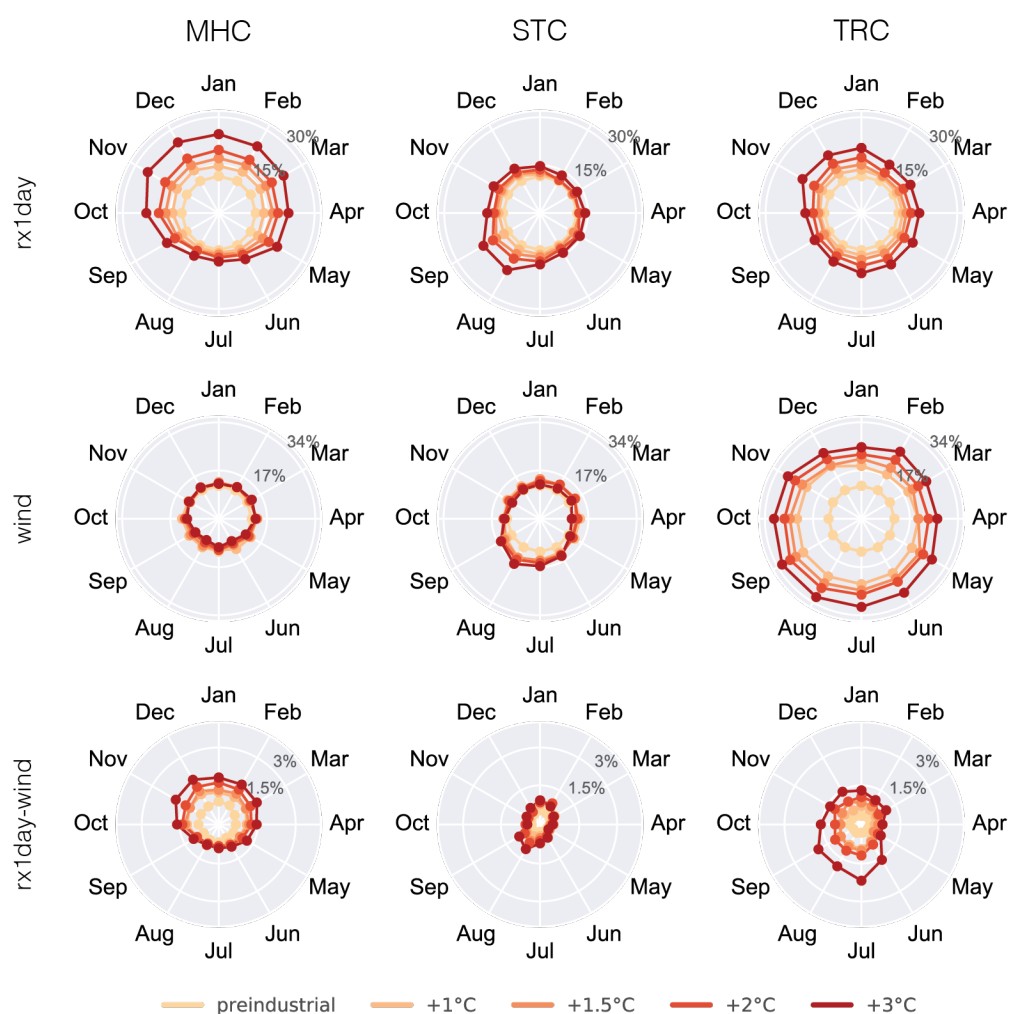

**Figure 5.** Same as Figure 3 but for Rx1day, wind and concurrent Rx1day-wind events.

### 3.3. Hotspots of Changes in Individual and Concurrent Extremes

This section presents the potential hotspots that are prone to an increase in exposure to multiple hazards in a future climate (Figure 6). We performed this analysis for the four individual event types (Figure 6 [a]) and the two concurrent event types (Figure 6 [b]) at GWLs. The first row shows how many of the event types increased at least 20% relative to the preindustrial period, and the second row shows how many of the event types increased at least 100% (i.e., a doubling of the event frequency).

Considering the individual extremes with the lower threshold (20%; Figure 6[a], top row), two out of four individual extremes show increase across almost the entire globe - even at a GWL of 1°C. There are three countries that show an increase in all extremes at 3°C GWL, namely Mali, Colombia, and Peru. Many countries, including most of the South American countries, European countries, the United States, Canada, China, some countries in Central West and South Africa

display change in three individual extremes at 3°C GWL (Figure 6[a] and B1. [a]). For the higher threshold and +1°C of global warming, two out of four individual events already doubled pre-industrial levels for countries in north and north-eastern South America and countries located in the south of the Mediterranean Sea. This increase is projected to continue and affect more land area for higher GWLs. The most prominent hotspots of change are Ghana, the Republic of Congo, Cameroon, Ecuador, Venezuela, Belize, Nicaragua, Guyana, and Colombia where the common driver are the heatwave events (Figure 6[a] and B1. [a]).

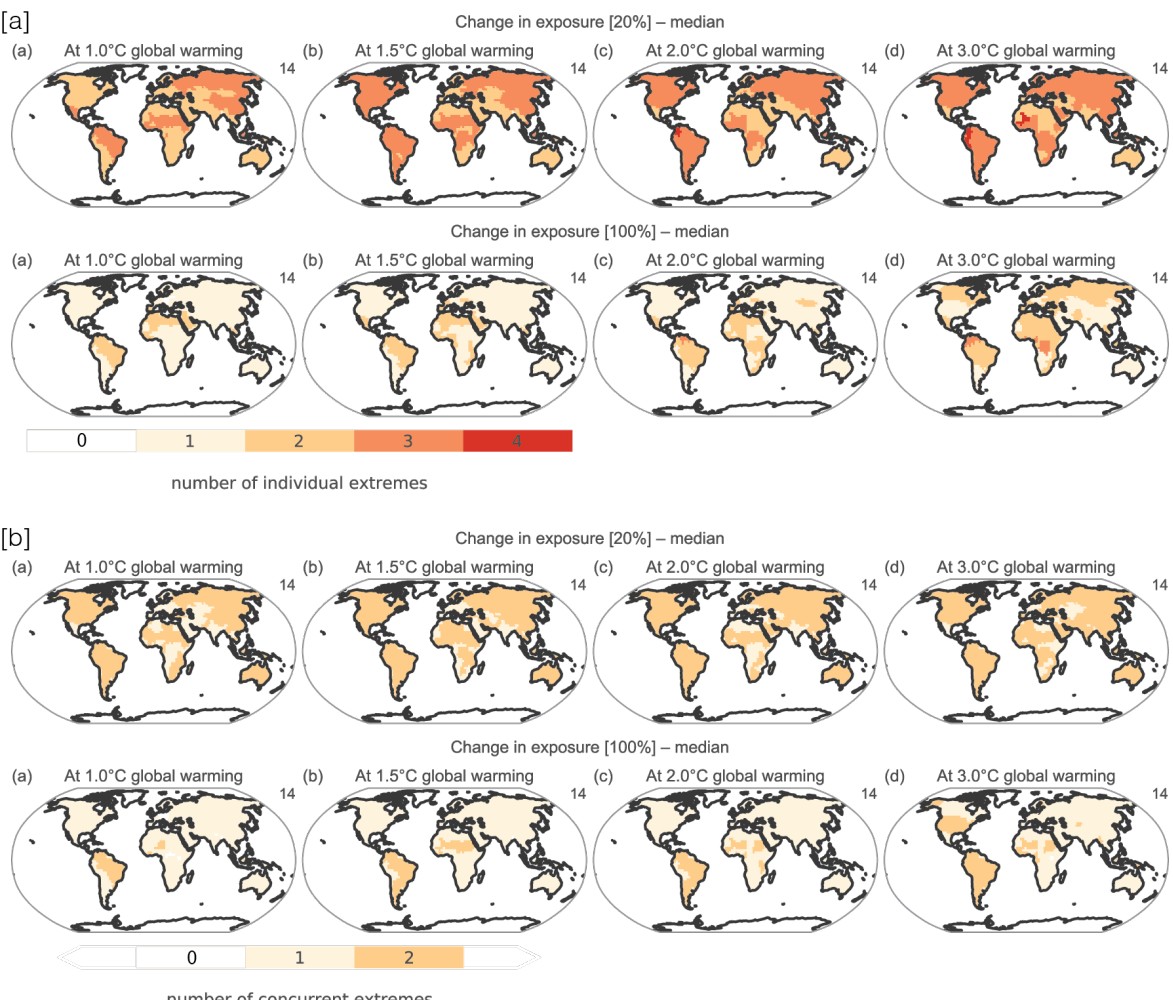

**Figure 6.** [a] Countries exposed to an increase in the event frequency for 1, 2, 3 or 4 individual extremes (heatwave, drought, Rx1day, wind) with relative increases over 20% (top row) and 100% (bottom row) with respect to the pre-industrial period [b] Countries exposed to 1 or 2 concurrent extremes (heatwave-drought, Rx1day-wind) with relative increases over 20% (top row) and 100% (bottom row) with respect to the pre-industrial period.

Both concurrent extreme pairs display a 20% increase at all GWLs in most countries except Mexico, India, some parts of Africa, Europe, and western Asia (Figure 6 [b] top row). This extends to almost the whole globe with higher GWLs. Some countries in South America and Western Africa already double the pre-industrial levels of heatwave-drought and Rx1day-wind frequency at 1°C warming (Figure 6 [b] bottom row). The regions where only one event doubles in frequency are mostly driven by heatwave-drought events (Figure B1. [b]). At 3°C of global warming both extreme pairs permanently double the pre-industrial levels for the United States, most countries in South America as well as some countries in Africa. At 3°C of global warming, Rx1day-wind events occurring in Mexico, western and central Europe, some countries surrounding the Mediterranean Sea do not contribute to 100% change, whereas for Kazakhstan and some countries in Africa its heatwave-drought events do not contribute to 100% change.

## 3.4. Population Exposure

Projected changes in individual and concurrent occurrences of heatwave, drought, Rx1day and wind events suggest a growing risk for population exposure across the globe. In addition, the global population is expected to continue its growth, further exacerbating the risk for human and natural systems. For example, SSP5 projects the average world population to grow from 7.29 billion in 2015 to its maximum in 2060 (8.6 billion) and decrease thereafter to about 7.4 billion people by 2100 - the lowest population size among SSPs (Jones and O'Neill, 2016, 2020). However, to estimate the population exposure on a country-by-country basis we use 2015 levels (7.33 billion) provided in the GPWv4 data. Thus, in this study, we don't consider increasing population from SSP5 but hold it constant at 2015 levels for several reasons. (i) Comparing GPWv4 with SSP5 projections suggest that the population in 2015 is 39 million people higher (7.29 billion) in SSP5 than in GWPv4 with an even higher discrepancy for 2020. (ii) Population projections are given for time periods while we report our results for global warming levels. Because each GCM reaches a warming level at a different period it would be difficult to assign a population number to the GWL. (iii) The projected population in SSP5 is strictly larger than in 2015, which suggests that our exposure based on 2015 population is conservative and gives a lower estimate.

Figure 7 shows the number of events per capita for 139 countries. The temporal span of this analysis is 20-years (20 years * 12 months = 240 time-steps) for each GWL. We multiply hazards (binary) (Figure B2) at each grid cell with the gridded population (Figure B3 (a)). We then sum all the values on the country level and divide it with the total population of that country (Figure B3 (b)). The obtained value is the number of events (or months) per capita in that specific country which cannot exceed 240. Using this approach allows us to consider the hazard at grid cells where population is not zero. Colours represent high model agreement (80% and above), and hatched areas represent low model agreement (less than 80%) in sign across models. In Figure 7, the first column represents the current (+1°C) number of events per capita and the second, third and fourth columns show the projected changes in the number of events at +1.5°C, +2°C and +3°C GWLs with respect to +1°C. Even when not taking the expected rise in the human population into account, increases in extremes alone are projected to increase the event number per capita in most countries. For +1°C of global warming, heatwave events range

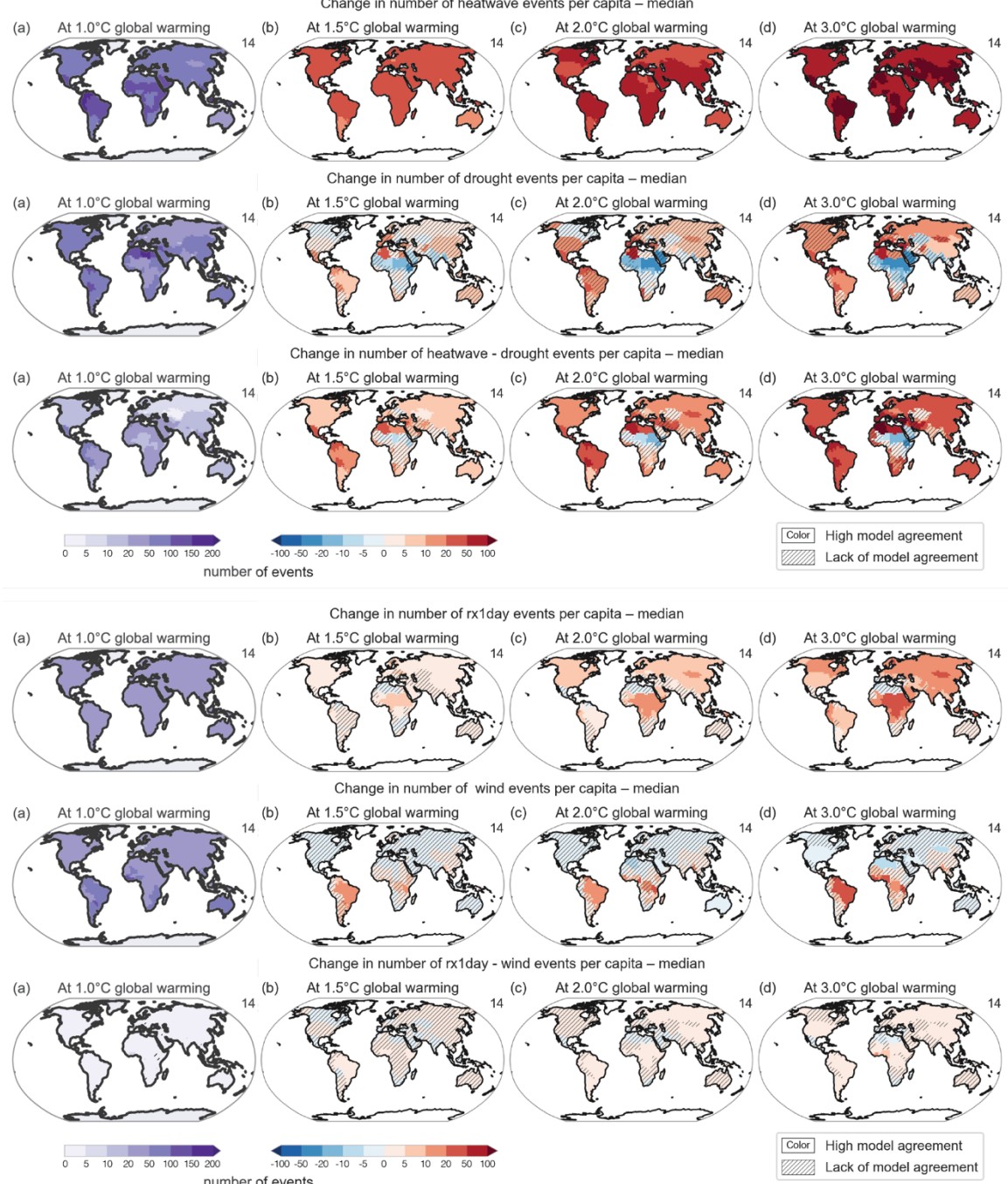

**Figure 7.** Number of individual and concurrent extremes per capita (a) at +1°C and (b, c, d) change at +1.5°C, +2°C and +3°C with respect to +1°C. 2015 population counts have been used for the analysis. Colours refer to high model agreement and hatched areas refer to lack of model agreement.

between ~34 and ~181 events per capita. The number of events per capita increases by ~10 to 51 events for +1.5°C, ~18 to 85 events for +2°C and ~45 to 146 events for +3°C GWL with high model agreement. The increase in number of events per capita for 80% of the countries is above 25 events, 51 events, and 86 events for +1.5°C, +2°C and +3°C GWLs with respect to +1°C. In case of drought, half of the countries indicate a continuous increase with higher GWLs up to ~78 more events. The most vulnerable countries for drought are the Mediterranean countries, China, some European countries, Mexico, and north-western countries of South America. Furthermore, the number of drought events per capita seems to be the least recurring event for some countries in Africa. Concurrent heatwave-drought events range between ~3 and ~88 events per capita across the globe for +1°C of global warming and it gradually increases for higher global warming levels for 82% of the countries with high model agreement. The number of events per capita increases gradually across the globe except for some countries in the African continent. The most dramatic increase is observed for countries in the Mediterranean Basin. The number of events tend to increase for all the countries in MHC, South America, and Australia more than 100 events per capita.

Individual Rx1day event numbers per capita are not very variable across the globe for +1°C of global warming (between ~23 and 41). The number of Rx1day events per capita is on the rise for higher global warming levels except Mediterranean countries, Australia, Mexico and north and south Africa countries, which some even show a small decrease (lack of model agreement). At +2°C of global warming, Rx1day events increase the most for tropical countries in the African continent. This increase continues for +3°C global warming almost everywhere across the globe. Wind extremes are increasing mostly for tropical countries including north-western countries of South America and some countries in central Africa. Most of the MHC and STC countries experience a decrease in the number of events per capita down to ~7 events for higher global warming levels (lack of model agreement). The number of events per capita for concurrent Rx1day and wind events are increasing in 41% of the countries. We observe the highest increase over the tropical countries in Africa up to ~10 more events.

**3.5. Non-event months**

Figure 8 shows the percentage of "normal" - non-event – months that countries experience, i.e., the percentage of months without any individual events studied in this paper (median of GCMs). We calculate the fraction of non-event months for each year (e.g., 6 normal months out of 12 corresponds to 0.5) over the 20-year period comprising the GWLs. We then take the mean of fractions and multiply with 100 to calculate the percentage of non-event months for each GWL. At preindustrial level, 60% of the months are normal meaning that there are ~7 normal months in every year across the globe. At the first glance, we see that with higher global warming levels, the percentage of normal months decreases gradually across the globe with some countries being more prone to the change. Independent from the frequency of events, all countries become a hotspots for individual extremes with increasing global warming. At current conditions, at ca. +1°C of global warming, 129 (out of 139) countries have 50% (6 months) or less normal months. 23 of these countries, mostly with a tropical climate,

have less than 20% (~2-3 month) normal months and 2 countries have less than 10% normal months (shown with grey colour) meaning that there is either one or no single month without individual events. At +1.5°C of global warming, the percentage of normal months is less than 20% for 51 countries. 21 of these countries are projected to have less than 10% normal months. These countries are mostly located in tropical climates. At +2.0°C of global warming, 79 countries are projected to have less than 20% of normal months whereas almost half of these countries (37) are projected to experience

extreme events every month. In a +3.0°C world, 85 countries experience the above mentioned 4 individual events almost every month whereas non-event months are between 10-20% for 41 countries and 20-30% for 11 countries. These results show that a large number of countries will shift to near-permanent extreme conditions even at global warming levels consistent with the limits of the Paris Agreement.

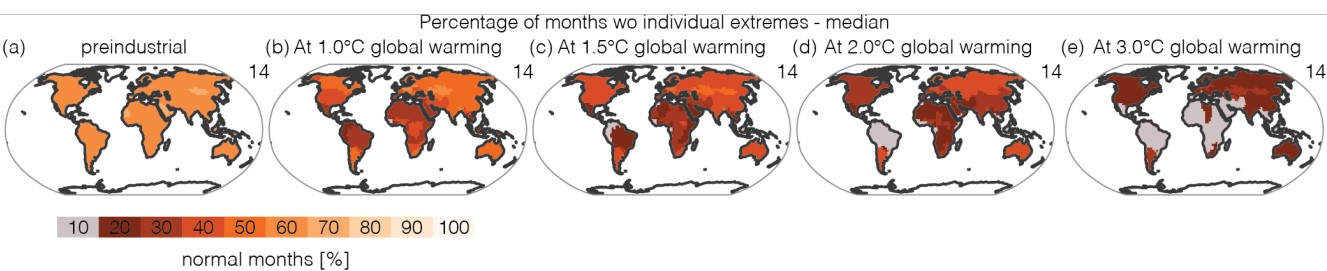

**Figure 8.** Percentage of non-event months (months without individual extremes) at warming levels (+1°C, +1.5°C, +2°C and +3°C).

## 4. Discussion

Our results highlight the increasing frequency of heatwaves, droughts, Rx1day and wind extremes with global mean warming. These findings, in particular the respective spatial patterns and increasing signals, are in accordance with the findings of the 6th Assessment Report of the Intergovernmental Panel on Climate Change (IPCC AR6). In the IPCC AR6 report, projected changes in annual maximum daily precipitation (Rx1day) and annual maximum temperature (TXx) indicate an increase over almost all land areas while soil moisture drought shows a heterogeneous pattern (Seneviratne et al., 2021).

Additionally, mean wind is expected to increase gradually in the 21st century in some tropical regions and decrease for the rest of the global land areas (Ranasinghe et al., 2021). In this work, individual heatwaves, droughts, Rx1day and wind extremes present consistent results with the above-mentioned indices. Increasing occurrence of these individual extremes can have important implications for natural and human systems. Therefore, the compatibility between the IPCC AR6 report and our results increases the confidence in our estimates of concurrent extremes that are associated with even more severe effects

than the respective individual extremes.

With higher global warming levels, we have seen a sharp increase in concurrent heatwave-drought events in three climate regions with the most dramatic increase in Northern Mid- and High Latitude Countries (MHC) followed by Subtropical

Countries (STC) (Figure 3a). As opposed to heatwave-drought events, Rx1day-wind events increase the most in Tropical Countries (TRC) (Figure 3b). The frequency differences among regions can be explained by varying climatic regimes. For instance, STC is more affected by warm-dry conditions than TRC because arid climate zones have more climate variability than equatorial climate zones. Another reason behind the frequency differences across regions can be the underlying dynamical and thermodynamic processes such as atmospheric circulation and teleconnection patterns. For example, compound droughts in Amazon are associated mainly with El Niño Southern Oscillation (ENSO) (Singh et al., 2021) and wet and windy extremes in north-western Europe are associated with the positive phase of the North Atlantic Oscillation (NAO) (De Luca et al., 2020). These findings correspond to regional findings in our analysis. Some studies have found that polar amplification weakens the north-south temperature gradient and warms up the cold extremes in mid- and high latitudes (Holmes et al., 2016; Gross et al., 2020), which is perhaps why MHC has prevailing heatwave-drought conditions. Another important thermodynamic process that can amplify temperature extremes is the lapse rate feedback mechanism. This mechanism increases temperature extremes in mid-high latitudes, while it decreases temperature extremes in tropics (Seneviratne et al., 2021). This direct influence on temperature extremes can be an indirect influence on precipitation extremes by altering the circulation patterns (dynamic processes) (Sillmann et al., 2017). Another key mechanism responsible for frequency increase can be the interaction between land and atmosphere (Seneviratne et al., 2010). Lack of moisture during droughts limits land evaporation which leads to an increase in sensible heat and in turn increases temperatures (Chiang et al., 2018). Furthermore, the change in moisture sources and sinks due to the future increases in greenhouse gas forcing will likely alter the hydrologic cycle (Batibeniz et al., 2020b) and such changes will likely intensify the land-atmosphere feedback mechanism causing concurrent warm and dry conditions. These explain why we found an increase in droughts occurring together with heatwaves in the projection period. Additionally, the enhancement of the concurrent very hot-dry warm seasons in many regions have also been linked with increasing dependence between temperature and precipitation associated with global warming (Zscheischler and Seneviratne, 2017). Moreover, it has been found that future occurrences of compound hot–dry events over land are connected with the variations in precipitation trends (Bevacqua et al., 2022). Our analyses show similar dipolar responses between heatwave-drought events and Rx1day-wind events in some countries.

Our results highlight the positive trend both in individual and concurrent events with higher global warming levels. The probability of occurrence of compound extremes is much lower than individual extremes by definition. Our results showcase this as individual events overall increase more than concurrent events. However, the multivariate structure of heatwave-drought and Rx1day-wind events change in the future across all climate regions. The interchangeable relationship between individual and concurrent events can be a sign of distributional changes in mean climate. In any case, increasing frequency decreases the number of non-event months without any individual extreme (Figure 8) which leaves less and less time for adaptation and recovery. Additionally, timing analysis indicates either abrupt increases or shifts in individual and concurrent extremes (Figure 4, 5). The inhomogeneous increases in frequency and changes in timing pose a risk for different sectors

such as agriculture, tourism, health. These changes may serve as a red flag for countries with an economy depending on these sectors.

Our analysis indicates that exposure to multivariate extremes is on the rise across the globe. For some countries, there is a dipolar pattern between exposures to heatwave-drought events and Rx1day-wind events. While the Mediterranean countries, Southern Africa and Mexico have an increase (decrease) in heatwave-drought (Rx1day-wind) events, Central Africa and Arabian Peninsula have a decrease (increase). Amazonia, Southern Africa, Sahel, India, and Southeast Asia have been projected as a hotspot for increasing temperatures and are the most vulnerable regions to extreme events (Bathiany et al., 2018). We find similar regional responses to increasing global warming levels. Low-income countries have been found to be more economically vulnerable to weather and climate extremes than rich countries (Jones and Olken, 2010; Dell et al., 2012, 2014). Therefore, these highly populated vulnerable countries that are prone to the largest changes in multi-hazard exposures could potentially be at larger risk.

The damage that extreme events cause is not only related to the frequency, severity or magnitude of the events but also to socioeconomic factors (Botzen et al., 2010; Jahn, 2015; Frame et al., 2020; IPCC, 2021) such as land use, income, education, employment and community safety. Different economic and social structures will alter the adaptive capacity to climate change. This makes it difficult to disassociate climate-related hazards from socioeconomic factors. Even so, assuming that projected future changes will take place in a world with a society and economy similar to today would help to understand the relative impacts of climate change on exposure. However, the global population is currently growing at a rate of around 1.1% per year, with the majority of this growth occurring in developing countries (Roser et al., 2013). The population living in the urban extent of Europe in 2015 is projected to increase more than 5% by 2050 (United Nations et al., 2019) and SSP population projections also estimate an increase in population (Jones and O'Neill, 2016). The distribution of population growth across different regions and demographic groups can vary, therefore, using population projections to investigate the human contribution to the change in exposure could help understand future risks more (Batibeniz et al., 2020a; Mukherjee et al., 2021). Our results provide evidence for an already existing vulnerability that may further increase in regions where extreme events will become more frequent due to climate change.

## 5. Conclusions

Investigating future changes in impactful individual and concurrent extremes is important to prepare for future climate risks. In this study, we have investigated the current state (~1°C of global warming) and projected change of individual and concurrent occurrences of heatwave, drought, Rx1day, wind events at global warming levels (GWLs) of +1.5°C, +2°C and +3°C relative to the pre-industrial period on the level of countries and climatic macro regions. Projections as function of GWLs provide useful information for stakeholders in the context of the Paris Agreement, which has set a limit for global warming stabilization "well below 2°C" and an aim to pursue efforts to limit global warming to 1.5°C (UNFCCC, 2015).

Analyses of simulations from 14 CMIP6 global circulation models allowed us to gain a robust understanding of extremes in current and future climate.

Our results indicate that all climate regions are under the increasing influence of concurrent hot-dry (heatwave-drought) events and Rx1day-wind events with higher GWLs. Even though this change is more substantial for heatwave-drought events, Rx1day-wind events are also on the rise. However, the order of the increase of events across regions shows a clear contrast. For heatwave-drought events, the increase is largest in Northern Mid- and High Latitude Countries (MHC), followed by Subtropical Countries (STC), and Tropical countries (TRC), whereas for Rx1day-wind events the order is the

opposite. While heatwave-drought events increased substantially, Rx1day-wind events increased less in MHC and STC. However, in TRC the increasing rate of heatwave-drought and Rx1day-wind events is similar, indicating the less variable climate in TRC. Isolated events are on the rise for heatwave and Rx1day events, whereas are decreasing for drought and wind events meaning that towards a warmer world, drought (wind) events are projected to co-occur with heatwave (Rx1day) events rather than occurring solely.

Our results also highlight the important timing shifts in the occurrence of individual and concurrent extremes in the future climate. Individual extreme events increase inhomogeneously across months leading to unprecedented frequency increases in some months in the future. Another important highlight of our study is increasing human exposure to concurrent extremes even without considering the expected rise in the human population. With higher GWLs, number of events per capita increases continuously in 53 countries for Rx1day-wind events, whereas this is valid for twice the number of countries for

heatwave-drought events. Our results also suggest non-event months are gradually decreasing for countries and that 85 countries will experience individual events nearly every month (i.e., less than 10% of non-event months) in a +3ºC warmer world. But this also affects several countries at 1.5ºC (21 countries) or 2ºC of global warming (37 countries). This shows that a large number of countries will shift to near-permanent extreme conditions (less than 10% of non-event months) even at global warming levels consistent with the limits of the Paris Agreement. Furthermore, our results suggest that there is a

prevailing increase in frequency, shifts in timing of concurrent extremes from +1.5ºC to +2.0ºC of global warming, thus exacerbating human exposure to these extremes with increasing global warming.

Despite many robust findings of our study, which are consistent with past assessments (Seneviratne et al., 2021) but also providing some new insights on the projected changes in extremes with increasing global warming, many sources of uncertainty need to be emphasized. This study relies on climate model simulations for both past and projected changes in

climate extremes. For historical changes, observational analyses could complement the provided results, but given the difficulty of investigating extreme events statistically due to their rare nature, climate models have been widely used for historical analyses in the literature (Sillmann et al., 2017; Miralles et al., 2019) using both regional and global climate models (Zhu and Yang, 2020; Zhu et al., 2020; Srivastava et al., 2020; Krishnan and Bhaskaran, 2020). We focus here on global simulations of standard resolution, which can be a limitation in regions of steep terrain. Indeed, high-resolution

regional models have been utilized especially for replication of wind and precipitation extremes at regions with complex local features (Coppola et al., 2021; Outten and Sobolowski, 2021; Reale et al., 2021; Stocchi et al., 2022), while global

climate models are often used to investigate the relationship between land surface conditions and extreme statistics (Seneviratne et al., 2013; Hauser et al., 2016; Rasmijn et al., 2018). However, the robust, large-scale investigation of extremes requires global model simulations with standard resolution, which often have lower computational cost compared to high-resolution global simulations and allow to obtain global statistical information compared to regional high-resolution simulations. Despite remaining uncertainties related to model deficiencies in some physical processes, natural variability (Wilcox and Donner, 2007; Rossow et al., 2013; Pfahl et al., 2017) and feedback mechanisms (Orlowsky and Seneviratne, 2013; Mueller and Seneviratne, 2014), CMIP6 is widely regarded as one of the most comprehensive and reliable sources for global information on climate change and is used in many extreme studies. Additionally, these models have a higher resolution, mostly higher climate sensitivity and produce better replication of physical, chemical, and biological processes, compared to CMIP5 (Coupled Model Intercomparison Project 5) used in IPCC AR5 (IPCC, 2021).

In conclusion, this study highlights the increasing occurrence of several single and compound extreme events with increasing global warming, with major increases in affected countries and human exposure even at levels of global warming consistent with the limits of the Paris Agreement. In particular, a substantial fraction of countries would be near permanently affected by extreme events already at 1.5°C, and even more so at 2°C and 3°C of global warming. The identified unprecedented changes in frequency and timing of extreme events would lead to an elevated risk for the environment and society across the globe. Therefore, our results suggest an urgent need for concrete actions to mitigate the current greenhouse gas emissions.

## Appendix A

**Table A1.** The list of CMIP6 GCMs

| No | GCM Name | Resolution | Ensemble |
|---|---|---|---|
| 1 | CMCC-CM2-SR5 | native atmosphere regular grid 1-degree 288 x 192 longitude/latitude | r1i1p1 |
| 2 | CMCC-ESM2 | native atmosphere regular grid 1deg; 288 x 192 longitude/latitude; 30 levels; top at ~2 hPa | r1i1p1 |
| 3 | EC-Earth3 | TL255, linearly reduced Gaussian grid equivalent to 512 x 256 longitude/latitude; 91 levels; top level 0.01 hPa | r1i1p1 |
| 4 | GFDL-CM4 | Cubed-sphere (c96) – 1-degree nominal horizontal resolution; 360 x 180 longitude/latitude; 33 levels; top level 1 hPa | r1i1p1 |
| 5 | HadGEM3-GC31-LL | native N96 grid; 192 x 144 longitude/latitude; 85 levels; top level 85 km | r1i1p1 |
| 6 | HadGEM3-GC31-MM | native N216 grid; 432 x 324 longitude/latitude; 85 levels; top level 85 km | r1i1p1 |
| 7 | INM-CM4-8 | gs2x1.5<br>2x1.5; 180 x 120 longitude/latitude; 21 levels; top level sigma = 0.01 | r1i1p1 |
| 8 | INM-CM5-0 | gs2x1.5<br>2x1.5; 180 x 120 longitude/latitude; 73 levels; top level sigma = 0.0002 | r1i1p1 |
| 9 | IPSL-CM6A-LR | LMDZ grid NPv6, N96; 144 x 143 longitude/latitude; 79 levels; top level 40000 m | r1i1p1 |
| 10 | MIROC6 | native atmosphere T85 Gaussian grid T85; 256 x 128 longitude/latitude; 81 levels; top level 0.004 hPa | r1i1p1 |
| 11 | MPI-ESM1-2-HR | spectral T127; 384 x 192 longitude/latitude; 95 levels; top level 0.01 hPa | r1i1p1 |
| 12 | MPI-ESM1-2-LR | spectral T63; 192 x 96 longitude/latitude; 47 levels; top level 0.01 hPa | r1i1p1 |
| 13 | MRI-ESM2-0 | native atmosphere TL159 gaussian grid (160x320 lat x lon)<br>TL159; 320 x 160 longitude/latitude; 80 levels; top level 0.01 hPa | r1i1p1 |
| 14 | UKESM1-0-LL | finite-volume grid with 1.9x2.5 degree lat/lon resolution<br>2 degree resolution; 144 x 96; 32 levels; top level 3 mb | r1i1p1 |

## Appendix B

[a]

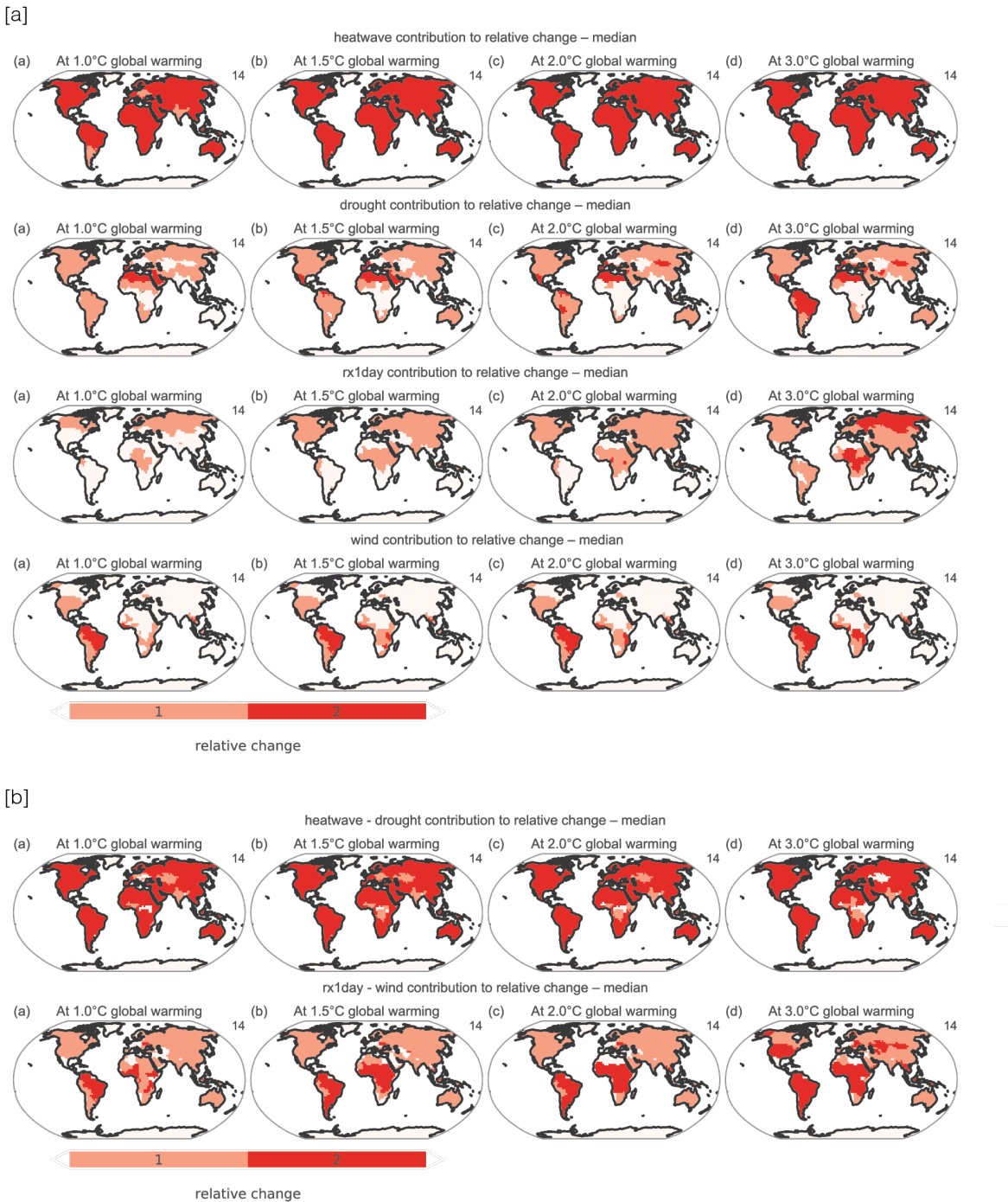

**Figure B1.** 20% relative change (shown with 1) and 100% relative change (shown with 2) of each individual extreme [a] and concurrent extreme [b].

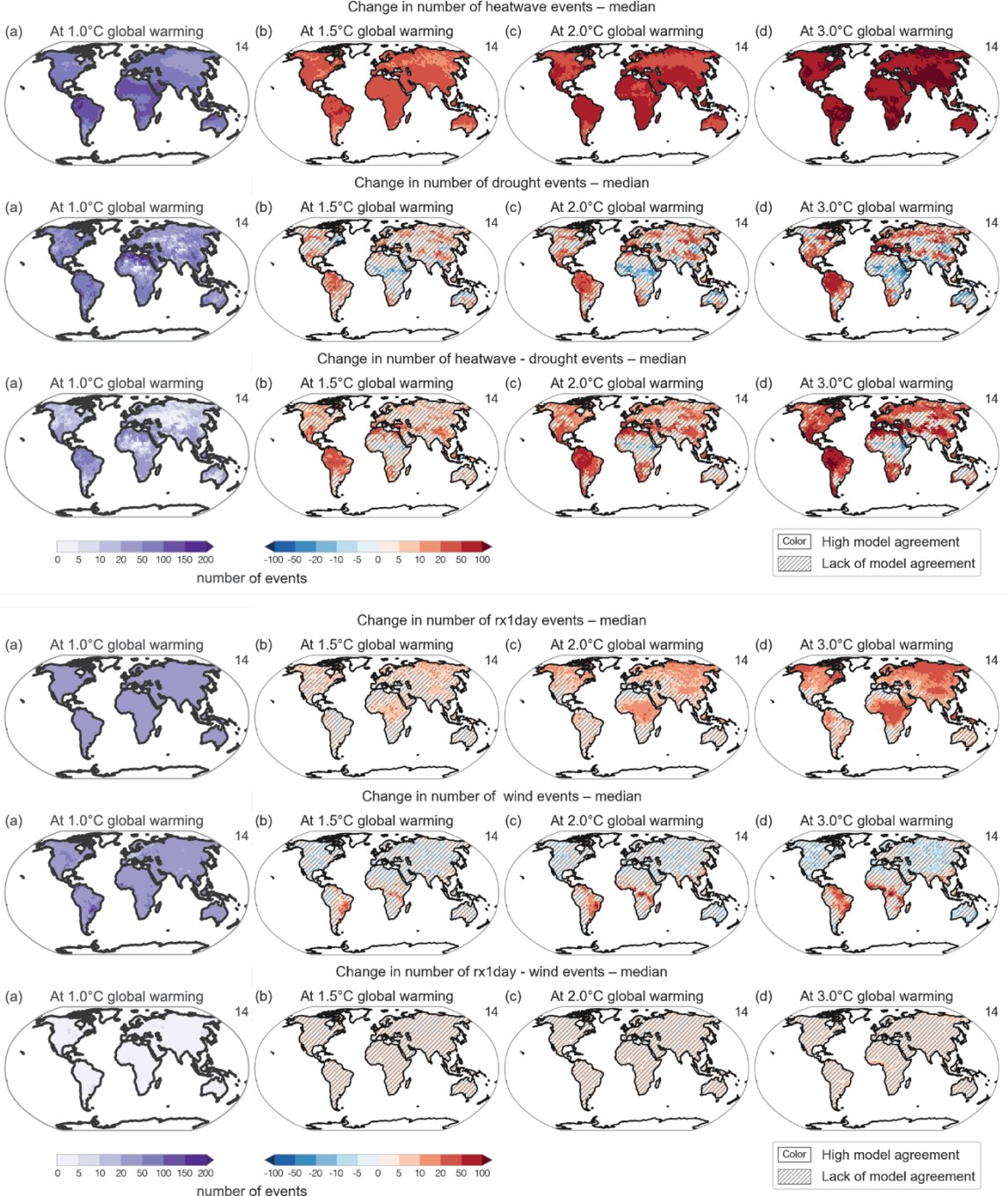

**Figure B2.** Number of individual and concurrent extremes (a) at +1°C and (b, c, d) change at +1.5°C, +2°C and +3°C with respect to +1°C. Colours refer to high model agreement and hatched areas refer to lack of model agreement.

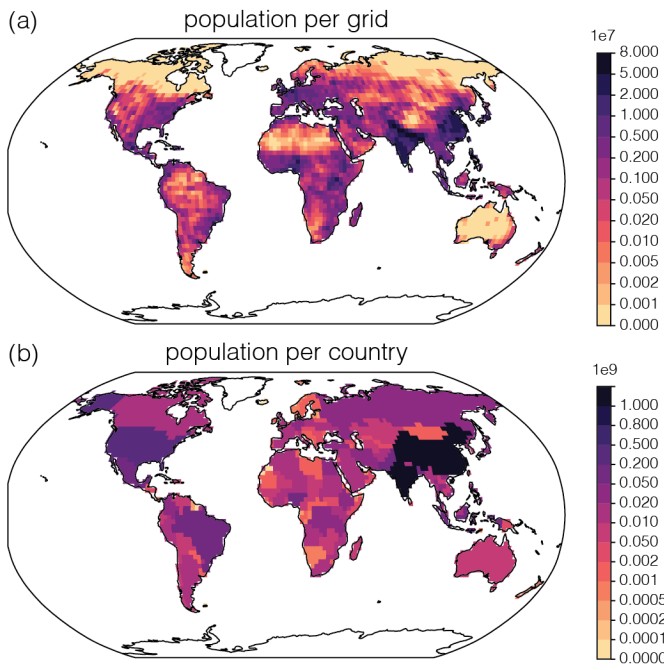

**Figure B3.** Population from GWPv4 at (a) 2.5º grid level and, (b) country level.

## 7. Data availability

GPWv4 used for population analysis are provided by NASA Socioeconomic Data and Applications Center (sedac) and is available at https://sedac.ciesin.columbia.edu/data/set/gpw-v4-population -count-rev11. Extreme indices have been generated using data archived on the ETH Zurich CMIP6 repository. Access to CMIP6 model outputs is also possible through different
Earth System Grid Federation (ESGF) data nodes.

## 8. Author contribution

FB, MH, and SIS planned the study; FB performed the analyses with support and guidance from MH and SIS; FB wrote the manuscript draft; MH and SIS reviewed and edited the manuscript. All authors were involved in discussions of the results
and streamlining the text.

## 9. Competing interests

The contact author has declared that neither they nor their co-authors have any competing interests.

## 10. Acknowledgements

This study was funded by the Swiss National Science Foundation (SNSF) through the Compound Events in a Changing Climate (CECC) (Grant agreement ID IZCOZ0_189941) project contributing to the European COST Action CA17109, "Understanding and modeling compound climate and weather events" (DAMOCLES).

We acknowledge the World Climate Research Program's Working Group on Coupled Modelling, which is responsible for the Coupled Model Intercomparison Project (CMIP), and we thank the climate modeling groups (listed in Appendix Table A1) for producing and making their model output available. Furthermore, we are indebted to Urs Beyerle, Lukas Brunner, and Ruth Lorenz for downloading and curating the CMIP6 data.

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
