# Peer review of "Countries most exposed to individual and compound extremes at different global warming levels"

_EGUsphere, 2022_

## Author Response (AR1)

We would like to thank the reviewers for their valuable comments. We have addressed all major and minor comments of the reviewers through appropriate changes and hope that the revised manuscript satisfies the reviewers' concerns.

The Response to the Reviewers file provides complete documentation of the changes made in response to each comment. The document is designed so that the changes that have been made in response to each comment can be immediately read and understood, independent of the other comments and responses. While this comprehensive comment-by-comment explanation requires some duplication of material throughout the document, our intention is that it helps to evaluate precisely how each comment has been addressed.

Reviewers' comments are shown in **bold**. The authors' response is shown in plain text. The text quoted from the manuscript is shown between quotation marks in *italics*.

**Summary of edits**
Here we would like to summarize the changes we have made in the manuscript. We have done considerable updates in the used methodology. For extreme wind event calculation, we now use daily maximum wind instead of daily wind and for drought events' calculation we now use daily data instead of monthly soil moisture data. We also changed our definition of concurrent events, and only marked the months where the extreme pairs occurred on the same day, whereas before it was on the month level.
We would like to also note that we changed the methodology of extreme indices and percentile calculation. It is mentioned previously in literature that percentile based indices for climate change detection may create artificial jumps at the beginning and end of the reference period (Zhang et al., 2005). These discontinuities can lead to an artificial frequency increase outside the reference period (e.g., at warming levels). To overcome this problem, we used the bootstrap resampling procedure proposed by Zhang et al. (2005). Indeed, this procedure improved our results in terms of inhomogeneities occurring outside the reference period for heatwave, maximum precipitation, and wind extremes.
Additionally, in previous analysis warming levels included 30-years but now we have changed it into 20-years consistent with the chapter on weather and climate extremes of the IPCC AR6 (Seneviratne et al., 2021).
We have extensively edited the text for every section to address all the questions/concern of the reviewers.

**Reviewer #1**
**This is a valuable study confirming, and bringing nuance to, the mounting evidence that extreme events – including compound extremes – are becoming more likely in a warming climate. The analysis of individual (soil moisture drought, heatwave, extreme precipitation and wind) and their concurrence is indeed interesting, but the paper suffers from a great deal of ambiguity making it hard to follow. For example, it is key for the audience to understand the global warming level scenarios, but the provided description lacks the required details (from text: "The warming levels are defined as the first 30-year period where global mean temperature anomalies exceed the given temperature (e.g. +2.0°C)." Similarly, the abstract provides details of how more frequent extremes will be, but does not clarify compared to what. Furthermore, coherence of the text and relevance of the presented materials to the main topic of the paper – specifically in the introduction section – needs improvement. In the following, I provide specific comments that hopefully are useful as the authors revise their manuscript:**

**Abstract: Needs more specific details about the presented statistics. View this from the lens of a general audience that might not be climate scientists. Anyone should be able to understand this synopsis of the paper, and at this point the text is too vague. For example: in Line 18, more frequent compared to what? Also, it might be helpful to define the period used to calculate extreme events' stats associated with various warming levels.**

We would like to thank the reviewer for their overall positive evaluation of our manuscript. We have made substantial changes in the manuscript in the light of these comments and hope that these revisions have addressed all the major or minor concerns.

In response to the reviewer's concern about the warming levels, we now explained the global warming levels extensively with a figure and text in the methodology section, which we believe will help readers to understand global warming levels better. We have added the new text stating *"Warming levels are 20-years periods unique to each model due to different climate sensitivity and internal variability. The warming levels are defined as the first 20-year period where global mean temperature anomalies exceed the given temperature (e.g., +2.0°C). We first calculate the annual average global temperature (Figure 2a). Then, we subtract the average global temperature of the pre-industrial period (1850-1900; reference period) from every year between 1850-2100 and take the 20-years running mean (Figure 2b). The first year a certain anomaly such as +1℃, +1.5℃, +2℃, and +3℃ is exceeded is the central year of the warming level period and the warming level period is obtained by subtracting ten and adding nine to the central year (Figure 2b, c; horizontal bars). For example, IPSL-CM6A-LR first exceeds +2℃ warming in 2036 so the period selected for this model is 2025-2044 (Figure 2c, red bar). On the other hand, MRI-ESM2-0 reaches +2℃ warming in 2040 and the period selected is 2029-2048 (Figure 2c, orange bar)."*

We agree with the reviewer's comment about the abstract, and we clarified the ambiguous sentences. We also believe understanding warming levels will remove the concern about the ambiguity in abstract regarding the frequency increase. We give all the frequency statistics with respect to the reference period where we can differentiate anthropogenic climate change from current (+1 ℃) and future global warming levels (+1.5 ℃, +2 ℃, +3 ℃). Only exception is the population exposure section where we use +1℃ as our comparison period for future warming levels. We have added the text stating *"It is now certain that human-induced climate change is increasing the incidence of extreme temperature, precipitation, and drought events globally. A critical aspect of these extremes is their potential concurrency that can result in substantial impacts on society and environmental systems. Therefore, quantifying concurrent extremes in current and projected climate is necessary to take measures and adapt to future challenges associated with such conditions. Here we investigate changes in individual and concurrent extremes in multi-model simulations of the 6th phase of the Coupled Model Intercomparison Project (CMIP6) for different global warming levels (GWLs). We focus on the individual and simultaneous occurrence of the extreme events, encompassing heatwaves, droughts, maximum 1-day precipitation (Rx1day) and extreme wind (wind), in the pre-industrial period (1850-1900; reference period), for approximately present conditions (1°C of global warming), and at three higher global warming levels (GWLs of +1.5°C, +2°C and +3°C). We focus our analysis on 139 countries and three climatic macro-regions: Northern Mid- and High Latitude Countries (MHC), Subtropical Countries (STC), and Tropical Countries (TRC). We find that, on a global scale, most individual extremes become more frequent and affect more land area for higher GWLs. Changes in frequency of individual heatwaves, droughts, Rx1day and extreme wind with higher GWLs cause shifts in timing and disproportional increases in frequency of concurrent events across different months and different regions. As a result,*

*concurrent occurrences of the investigated extremes become 2.0 to 9.6 times more frequent at +3°C of global warming compared to the pre-industrial period. At +3°C the most dramatic increase is identified for concurrent heatwave-drought events with a 9.6 times increase for MHC, a 8.4 times increase for STC, and a 6.8 times increase for TRC compared to the pre-industrial period. By contrast, Rx1day-wind events increased the most in TCR (5.3 times), followed by STC (2.3 times) and MHC (2.0 times) at +3°C with respect to the pre-industrial period. Based on the 2015 population, these frequency changes imply an increase in the number of concurrent heatwave-drought (Rx1day-wind) events per capita for 82% (41%) of countries. Our results also suggest that there are almost no time periods (on average zero or only one month per year) without heatwaves, droughts, Rx1day and extreme wind for 21 countries at +1.5°C of global warming, 37 countries at +2°C and 85 countries at +3°C, compared to 2 countries at 1°C of global warming. This shows that a large number of countries will shift to near-permanent extreme conditions even at global warming levels consistent with the limits of the Paris Agreement. Given the projected disproportional frequency increases and decreasing non-event months across GWLs, our results strongly emphasize the risks of uncurbed greenhouse gas emissions."*

**Introduction: There are much information that is not relevant to the paper. For example, you are not addressing vulnerability and adaptive capacity in this paper, and I wonder how paragraph starting in line 47 builds up the foundation for your research question? I would use this information in the discussion, but not introduction. Also, information presented in many paragraphs are not aligned with the topical sentence. For example, I don't know how naming specific countries in lines 40-42 can be helpful in this section. Furthermore, lines 42-45 are neither relevant to the study topic nor aligned with the topical sentence. In general, the introduction section can be sharpened to interest the audience.**

Thanks for the comment. We have revisited the introduction section by deleting the suggested paragraphs/information and by making extensive changes to sharpen our message. We are not sharing the revised introduction here due to its length, however, it is available within the provided pdf file.

**Lines 89-90 read: "we investigate here for the first time the human exposure to these concurrent extremes in addition to individual extremes." -> This sentence is not correct. See for example:**

**Concurrent: Liu, W., Sun, F., Feng, Y., Li, C., Chen, J., Sang, Y.F. and Zhang, Q., 2021. Increasing population exposure to global warm-season concurrent dry and hot extremes under different warming levels. Environmental Research Letters, 16(9), p.094002.**

**Individual: Alizadeh, M.R., Abatzoglou, J.T., Adamowski, J.F., Prestemon, J.P., Chittoori, B., Akbari Asanjan, A. and Sadegh, M., 2022. Increasing Heat Stress Inequality in a Warming Climate. Earth's Future, 10(2), p.e2021EF002488.**
**And many more studies, including some from the authors' group.**

We thank the reviewer for bringing these papers up. We were not aware of these papers. We now added the mentioned references to the revised manuscript and revised the last part of the introduction in the light of these comments. The revised last paragraph states *"Here we analyse changes in frequency and timing of climate-induced individual and concurrent extreme events, as well as the population exposure to these events. It is important to note that for a risk*

*assessment vulnerability would also have to be considered, but this lies beyond the scope of this study. Building on previous work on projected changes in compound extreme events and human exposure (Batibeniz et al., 2020; Lange et al., 2020; Chen et al., 2020; Mukherjee et al., 2021; Liu et al., 2021; Alizadeh et al., 2022; Das et al., 2022; Shen et al., 2022)(Batibeniz et al., 2020; Lange et al., 2020; Chen et al., 2020; Mukherjee et al., 2021; Liu et al., 2021; Alizadeh et al., 2022; Das et al., 2022; Shen et al., 2022), we investigate for the first time the human exposure to co-occurring extreme precipitation-wind events, in addition to co-occurring heatwave-drought events and individual extremes. We do so in a manner consistent with the 6th Assessment Report of the Intergovernmental Panel on Climate Change (IPCC AR6) framework by analysing the projections for different global warming levels (GWLs, +1°C, +1.5°C, +2°C and +3°C) relative to pre-industrial conditions on country and regional scales."*

**Sections 2.5. & 2.6. and across the manuscript: The definitions of individual extremes and compound extremes are confusing. I learned half-way through the manuscript that drought refers to a drought that is not concurrent with heatwave. The entire paper needs to be revisited to clarify what each of the extremes (individual or compound/concurrent) refers to. Also, and importantly, the temporal resolution of extremes needs clarification. I understand how droughts are monthly and heatwaves are daily, and how the authors label a month as observing concurrent drought-heatwaves, but I struggle with how concurrent extreme precipitation and wind is defined. As it stands, it seems like a month that has one of each event is labeled as observing concurrent extreme precipitation and wind, which is not correct (at least in my opinion). The impact of extreme precipitation and wind are most pronounced at the daily scale (or even hourly, but let's stick to daily), and that should be the temporal scale of the analysis. If one occurs at the beginning of a month and the other occurs at the end of the month, that month should not be tagged as having observed a concurrent extreme precipitation and wind.**

We appreciate this suggestion. We have revisited and clarified the definition of extremes. In response to reviewer's concern on concurrent event definition; we have changed our definition of concurrent events, and we have only marked the months where there the extreme pairs occurred on the same day. The text explaining the calculation procedure is *"We define concurrent events as events that occur on the same day in a month and affect the same location. We assess two types of concurrent events: combined heatwave and drought events as well as Rx1day and extreme wind events. Thus, if a specific month experiences two individual events on the same day, it is marked as "event month", for that grid cell and month. For example, if there is a drought event occurring on the same days with a heatwave event regardless of the number of concurrent events, we mark that month as an "event month" otherwise "non-event month".*

**Line 179 reads "Drought events, on the other hand, tend to decrease for higher GWLs in MHC and STC." This is confusing/misleading for the reason mentioned above. Similarly, lines 223-224 read "Interestingly, STC sees a small decrease in individual drought events in most months for 3°C warming." Which is again misleading due to the definition and lack of clarity of the text.**

Thanks for pointing this out. We now edited the text in the light of this comment. We rephrased some sentences to clarify the text and instead of individual we used isolated events to avoid confusion.

**On a technical note, how reliable are Rx1day simulations/projections? How about wind?**

Thanks for this comment. We have now added new text that briefly describes the advantages and limitations of CMIP6. The revised text states *"Despite many robust findings of our study, which are consistent with past assessments (Seneviratne et al., 2021) but also providing some new insights on the projected changes in extremes with increasing global warming, many sources of uncertainty need to be emphasized. This study relies on climate model simulations for both past and projected changes in climate extremes. For historical changes, observational analyses could complement the provided results, but given the difficulty of investigating extreme events statistically due to their rare nature, climate models have been widely used for historical analyses in the literature (Sillmann et al., 2017; Miralles et al., 2019) using both regional and global climate models(Zhu and Yang, 2020; Zhu et al., 2020; Srivastava et al., 2020; Krishnan and Bhaskaran, 2020). We focus here on global simulations of standard resolution, which can be a limitation in regions of steep terrain. Indeed, high-resolution regional models have been utilized especially for replication of wind and precipitation extremes at regions with complex local features (Coppola et al., 2021; Outten and Sobolowski, 2021; Reale et al., 2021; Stocchi et al., 2022), while global climate models are often used to investigate the relationship between land surface conditions and extreme statistics (Seneviratne et al., 2013; Hauser et al., 2016; Rasmijn et al., 2018). However, the robust, large-scale investigation of extremes requires global model simulations with standard resolution, which often have lower computational cost compared to high-resolution global simulations and allow to obtain global statistical information compared to regional high-resolution simulations. Despite remaining uncertainties related to model deficiencies in some physical processes, natural variability (Wilcox and Donner, 2007; Rossow et al., 2013; Pfahl et al., 2017) and feedback mechanisms (Orlowsky and Seneviratne, 2013; Mueller and Seneviratne, 2014), CMIP6 is widely regarded as one of the most comprehensive and reliable sources for global information on climate change and is used in many extreme studies. Additionally, these models have a higher resolution, mostly higher climate sensitivity and produce better replication of physical, chemical, and biological processes, compared to CMIP5 (Coupled Model Intercomparison Project 5) used in IPCC AR5 (IPCC, 2021)."*

**In general, it would be helpful to discuss why certain temporal and spatial patterns are projected. For example, lines 226-228 (among others) can benefit from this.**

We have repeated the analyses and revised the whole text.

**I struggled to understand how population projections are used in this analysis. It seems like the 2015 population data was used only. Please clarify the text.**

Thanks for the insight. We revisited and clarified the text for the population counts section. Indeed, we only use 2015 Gridded Population of the World version 4 (GPWv4) data to calculate the population exposure in section 3.4. Therefore, we removed information related with population projections of shared socioeconomic pathways (SSP5) to avoid confusion.

**I also struggled to understand how the number of events per person on the country basis was calculated. Are you calculating exposure (person-days) and dividing it by the country population? In any case, please clarify. Also clarify what you mean by certain counts of extremes per person. What temporal span does this refer to? Annual? Decadal? 30 years?**

The temporal span of this analysis is 20-years (20*12=240 time-steps). We multiply the population with hazards (binary) at each grid cell. Sum all the values on the country level and divide it with the total population of that country. The obtained value is the number of events (or months) per person in that specific country which cannot be more than 240. We have now explained it in more detail in the section 3.4 Population Exposure.

**Lines 303-304 read "The number of events per person increases gradually across the globe except tropical countries in the African continent and India." While it might be beyond the scope of this paper, it would be interesting to discuss how the decomposition of population dynamics (if it is considered here) and count (e.g., country population) vs extremes frequency trends contributed to these statistics.**

Thanks for the suggestion. We have included this into the discussion. The text states *"The damage that extreme events cause is not only related to the frequency, severity or magnitude of the events but also to socioeconomic factors (Botzen et al., 2010; Jahn, 2015; Frame et al., 2020; IPCC, 2021) such as land use, income, education, employment and community safety. Different economic and social structures will alter the adaptive capacity to climate change. This makes it difficult to disassociate climate-related hazards from socioeconomic factors. Even so, assuming that projected future changes will take place in a world with a society and economy similar to today would help to understand the relative impacts of climate change on exposure. However, the global population is currently growing at a rate of around 1.1% per year, with the majority of this growth occurring in developing countries (Roser et al., 2013). The population living in the urban extent of Europe in 2015 is projected to increase more than 5% by 2050 (United Nations et al., 2019) and SSP population projections also estimate an increase in population (Jones and O'Neill, 2016). The distribution of population growth across different regions and demographic groups can vary, therefore, using population projections to investigate the human contribution to the change in exposure could help understand future risks more (Batibeniz et al., 2020; Mukherjee et al., 2021). Our results provide evidence for an already existing vulnerability that may further increase in regions where extreme events will become more frequent due to climate change."*

**Are you using multi-model mean or median? Line 173 says "mean" and line 327 says "median"**

We are using multi-model mean for Venn diagrams and multi-model median for the rest of the analysis. In Venn diagrams, we calculate (A∩B), A-(A∩B), and B-(A∩B) (sets) for each model separately. To avoid showing different shares from different models for each set, we illustrate the mean.

**Lines 339-341 read "Northern parts of South America especially Bolivia, Chile, Paraguay and Brazil, South Africa, the United States of America, Australia and Mexico are also very vulnerable to this change.". Confusing sentence. The value of naming specific countries in some context and referring to regions in other contexts is not clear to me**

Thanks for the comment. We agree this is confusing. We have revised the whole section.

**Lines 395-396 read "Therefore, using population projections to investigate the human contribution to the change could help understand future risks more". Not clear**

Thanks for the insight. We have updated the paragraph this sentence was in. The text states *"The damage that extreme events cause is not only related to the frequency, severity or magnitude of the events but also to socioeconomic factors (Botzen et al., 2010; Jahn, 2015; Frame et al., 2020; IPCC, 2021) such as land use, income, education, employment and community safety. Different economic and social structures will alter the adaptive capacity to climate change. This makes it difficult to disassociate climate-related hazards from socioeconomic factors. Even so, assuming that projected future changes will take place in a world with a society and economy similar to today would help to understand the relative impacts of climate change on exposure. However, the global population is currently growing at a rate of around 1.1% per year, with the majority of this growth occurring in developing countries (Roser et al., 2013). The population living in the urban extent of Europe in 2015 is projected to increase more than 5% by 2050 (United Nations et al., 2019) and SSP population projections also estimate an increase in population (Jones and O'Neill, 2016). The distribution of population growth across different regions and demographic groups can vary, therefore, using population projections to investigate the human contribution to the change in exposure could help understand future risks more (Batibeniz et al., 2020; Mukherjee et al., 2021). Our results provide evidence for an already existing vulnerability that may further increase in regions where extreme events will become more frequent due to climate change."*

**Minor comments:**
**Line 39: "and"->"that"**

Thanks. We fixed it.

**Compound extremes in this paper refers to concurrent extremes, if I understand correctly. It might be helpful to be specific throughout the paper.**

Thanks for the insight. We are now using concurrent extremes throughout the paper.

**Reviewer #2**
**This is an interesting and important study focussing on changes in four individual and two concurrent extremes at different warming levels (GWLs) with reference to the pre-industrial level (1850-1900) based on multiple CMIP6 models. These findings are important to understand the changes in frequency of extremes at present and probable warming levels in the future. However, I have two major concerns: 1) the study involves statistical analysis but offers very little on physical linkages in model processes and extreme weather events, and 2) the manuscript lacks clarity in description of the adopted methods. On a broader context, there are several studies that are coming out in the recent times that are merely the outcome of the CMIP6 models represented in terms of charts and maps, offering very little on science and understanding. I hope the authors will add discussions and usability beyond that.**

We would like to thank the reviewer for their overall positive evaluation of our manuscript. We have made substantial changes in the manuscript to eliminate the concerns of the reviewers. We now explained our methodology with more details/illustrations and improved our discussion in a way to include more physical linkages.

**Further, the study starts discussing high vulnerability of tropical countries to climate extremes. Yet, it limits itself to exposure of extremes to the population, ignoring other**

**indicators relevant to the individual extremes. While this may be beyond the scope of the study, I would suggest rewriting the introduction part for better communication.**

Thank you for your insight. We have restructured our introduction in the light of this comment. Because of its length, we recommend reading the pdf of our manuscript.

**Other points:**

**1- There has been discussion on population data from different sources and interpolating data from 2000-2100; however, later on, the population of 2015 is employed for the determining exposure. Hence, the significance of lines 116-118 is difficult to understand.**

Thanks for pointing this out. We revisited and clarified the text for the population counts section. Indeed, we only use 2015 Gridded Population of the World version 4 (GPWv4) data to calculate the population exposure in section 3.4. Therefore, we decided to remove information related with population projections of shared socioeconomic pathways (SSP5) to avoid confusion.

**2- In addition to the reference provided for the methodology to select data for different warming levels, some details in addition to Line 135-137 would clarify the audience.**

We thank the reviewer for this suggestion. In response to the reviewer's concern about the warming levels, we have now explained the global warming levels extensively with a figure and text in the methodology section, which we believe will help readers to understand global warming levels better. We have added the new text stating *"Warming levels are 20-years periods unique to each model due to different climate sensitivity and internal variability. The warming levels are defined as the first 20-year period where global mean temperature anomalies exceed the given temperature (e.g., +2.0°C). We first calculate the annual average global temperature (Figure 2a). Then, we subtract the average global temperature of the pre-industrial period (1850-1900; reference period) from every year between 1850-2100 and take the 20-years running mean (Figure 2b). The first year a certain anomaly such as +1℃, +1.5℃, +2℃, and +3℃ is exceeded is the central year of the warming level period and the warming level period is obtained by subtracting ten and adding nine to the central year (Figure 2b, c; horizontal bars). For example, IPSL-CM6A-LR first exceeds +2℃ warming in 2036 so the period selected for this model is 2025-2044 (Figure 2c, red bar). On the other hand, MRI-ESM2-0 reaches +2℃ warming in 2040 and the period selected is 2029-2048 (Figure 2c, orange bar)."*

**3- I also have some reservations on the monthly temporal scale, based on which concurrent extremes are determined here. I hope the authors have considered the timing of the events in a month, particularly for Rx1day and extreme wind, to declare the two events as concurrent. More details related to Line 163-167 are required to clarify concurrent extremes.**

In response to reviewer's concern on concurrent event definition; we have changed our definition of concurrent events, and we have only marked the months where there the extreme pairs occurred on the same day. The text explaining the calculation procedure is *"We define concurrent events as events that occur on the same day in a month and affect the same location. We assess two types of concurrent events: combined heatwave and drought events as well as*

*Rx1day and extreme wind events. Thus, if a specific month experiences two individual events on the same day, it is marked as "event month", for that grid cell and month. For example, if there is a drought event occurring on the same days with a heatwave event regardless of the number of concurrent events, we mark that month as an "event month" otherwise "non-event month".*

**4- I am unclear on the method adopted for the event fraction and frequency (Line 176-177) of extremes—an explanation of how to reach the particular fraction need to be added.**

Thanks for the comment. We revised the text to avoid this confusion. The frequency and fraction have the same meaning in the text. The pre-industrial period has 612 timesteps (51-years * 12 months) and global warming level periods have 240 timesteps (20-years * 12 months). Each timestep is either 1 or 0. We average these values over time and the resulting value is between 0 and 1 indicating a fraction of that period exposed to specific extremes. To ease understanding we multiply it with 100 and give it as a percentage.

**5- In Figure 5b(top row), the number of concurrent extremes for the Indian region changes from 2 at present GWL to 1 at 1.5°C and then changes further. How does the number of concurrent extremes change at 2°C and 3°C for that region? Also, the possible reason behind this need to be explained in the corresponding section.**

Thanks for pointing the problem out. We repeated all the analysis so the values for India changed and fixed. The previous values were caused due to plotting mistakes and they are now fixed.

References:

[revised manuscript text omitted]

---

## Author Response (AR2)

We would like to thank the reviewers for their valuable comments and finding our effort enough to decide the manuscript is ready for publication.

The Response to the Reviewers file provides complete documentation of the changes made in response to each comment. The document is designed so that the changes that have been made in response to each comment can be immediately read and understood, independent of the other comments and responses.

Reviewers' comments are shown in **bold**. The authors' response is shown in plain text. The text quoted from the manuscript is shown between quotation marks in *italics and new text is shown in green*.

**Anonymous Reviewer**

**The manuscript highlights the significance of concurrent extreme events under the effect of different global warming levels. Climate change induced several extreme climatic conditions to occur at the same time leading to severe damage. The manuscript also offers the regional impacts of different concurrent events: heatwave-drought is most in the mid-high latitude regions and Rx1day-wind speed in the tropics. The manuscript is well-written and organized, with a clever representation of the results. The authors replied extensively to the comments addressed by the previous reviewers and made considerable changes in the methodology, going from monthly to daily analyses. According to my review, the manuscript is suitable for publication. I suggest the following optional minor comments for the authors' consideration:**

We would like to thank the anonymous reviewer for her/his positive evaluation and considering our previous efforts. We have addressed the minor comments through appropriate changes and hope that the revised manuscript satisfies the reviewer's concerns.

**(1) In the abstract, add a sentence to define the two types of concurrent events before jumping into the results.**

We thank the reviewer for pointing this out. We have now edited the following sentence in abstract to make this clear to the reader. The new text states:

*"We focus on the individual and simultaneous occurrence of the extreme events, encompassing heatwaves, droughts, maximum 1-day precipitation (Rx1day), extreme wind (wind) as well as the compound events heatwave-drought and Rx1day-wind in the pre-industrial period (1850-1900; reference period), for approximately present conditions (1°C of global warming), and at three higher global warming levels (GWLs of +1.5°C, +2°C and +3°C)."*

**(2) It would be great to add the locations of the 139 countries you are considering in this analysis on a map to see their regional distribution. You can add it to Figure 01 if you prefer.**

We thank the reviewer for this comment. Current version of the figure has all 139 countries considered in the analysis.

**(3) Although the authors justified using global warming levels to align with the IPCC reports, the socio-economic pathways (SSPs) add the socio-economic perspective besides global warming. SSP5-8.5 is the only adopted socio-economic scenario. I would appreciate a few sentences explaining why you did not use more socio-economic scenarios. A probable explanation could be that you focus on the cause-effect relationship between increasing temperatures and concurrent extreme events. Build on sentences around L435.**

Thank you for your suggestion. We would like to kindly bring to your attention that our analysis did not involve the use of any SSP population projection. Instead, we used the GPWv4 dataset's 2015 population because the timing of each model reaching certain warming levels varied in the projection period. We believe that the paragraph in question provides a comprehensive explanation of the potential population growth in the future without emphasizing any particular SSP scenario. We considered your suggestion regarding adding information around the highlighted sentence (L435). However, we think doing so may disrupt the paragraph's flow. Nevertheless, we agree that this information needs to be communicated. Therefore, we have added this information to the population counts section. The new text now states:

*"In our population exposure analysis, we use gridded population counts retrieved from the Gridded Population of the World version 4 (GPWv4) dataset (Center for International Earth Science Information Network - CIESIN - Columbia University, 2018). The GPWv4 dataset provides population distributions at various grid resolutions. For our analysis, we use the 1° resolution data, which we transform into 2.5° grid resolution to match the resolution of the climate data. The GPWv4 dataset covers the period from 2000 to 2020 at 5-year intervals. However, we only use 2015 population counts in this paper as they are representative of the world population at +1°C of global warming. To investigate the effect of climate change, we keep the population fixed at 2015 levels for approximately 1°C of global warming while allowing the counts of climate events to change at GWLs. This approach enables us to examine the cause-effect relationship between increasing temperatures and projected changes in extreme events. Furthermore, using climate change projections and population distributions in combination allows us to investigate changes in exposure to climate extremes at the regional and country levels."*